# PM1: A Foundation Model Fusing Genotype, Phenotype, and Image for Precision Medicine

## Abstract

Precision medicine aims to personalize disease prevention, prediction, and diagnosis by leveraging genomic patient data. Although patient genomes provide valuable predictive insight, they cannot capture the full complexity of an individual's health. Integrating genomics with additional patient data modalities, such as clinical phenotypes and medical imaging, enables more accurate and comprehensive disease modeling. We introduce **PM1**, a multimodal foundation model trained on genomic data from 438,668 individuals linked to 3,421 clinical and lifestyle traits and 211,416 retinal fundus photographs drawn from the UK Biobank and EyePACS cohorts. PM1 couples modality-specific encoders with a transformer encoder trained with an information noise-contrastive estimation objective that fuses modalities into a joint latent space, plus generative modality decoders for cross-modal reconstruction and synthesis. A token-level masking schedule lets PM1 use participants with *any* subset of modalities (in UK Biobank only $\approx 6\%$ have all three), substantially expanding effective training data. Joint modeling of retinal images, clinical traits, and genomic data surpasses single-modality and multimodal baselines. PM1 enables cross-modal genotype inference, raises predictive performance for retinal diseases and systemic conditions, and supports conditioned single nucleotide polymorphism sequence and retinal image generation. As a group-level validation, a GWAS on PM1's image-conditioned fusion embeddings recovers genome-wide significant HERC2 pigmentation variants.

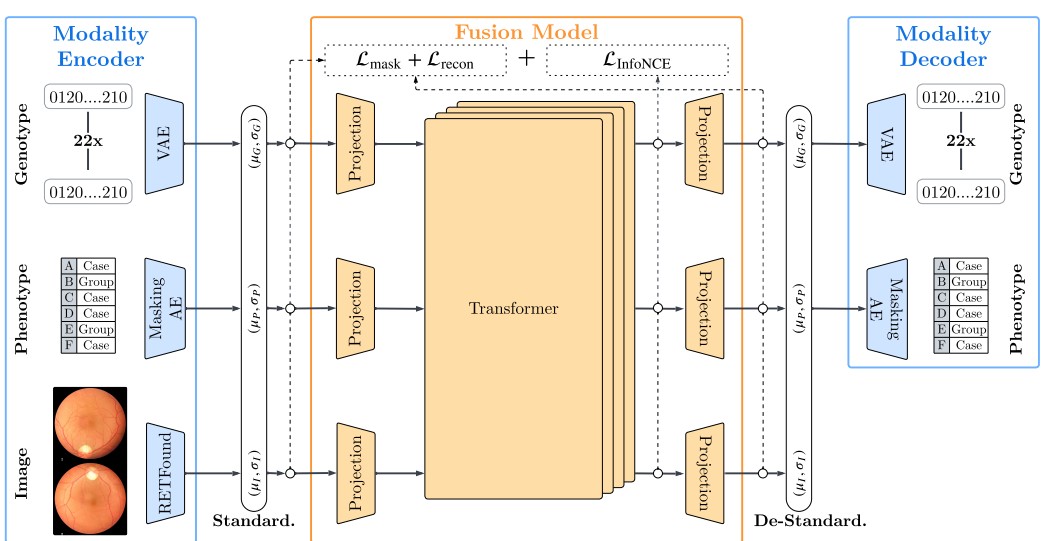

Figure 1: **Overview of PM1 Model Architecture**. Encoders for genotype, phenotypes, and retinal images emit tokens mapped to a shared space, fused by a transformer for alignment and prediction, with modality decoders reconstructing inputs and enabling synthesis when inputs are missing.

## 1 INTRODUCTION

Following the first demonstration of generalist multimodal biomedical foundation models in 2023 (Tu et al., 2024; Li et al., 2023; Zhang et al., 2023; Theodoris et al., 2023), mostly spanning clinical text and imaging data, these systems still omit the most heritable axis of human variation—whole-genome sequence—hence ignoring information that underlies virtually every complex trait. Integrating high-dimensional genomic data into multimodal models poses unique challenges: genomic sequences are both longer and more structured than typical language or image inputs, exhibiting strong correlations through linkage disequilibrium (LD). Meanwhile, biobank-scale resources like the UK Biobank (UKB) (Sudlow et al., 2015) provide an invaluable resource for studying the relationship between genomes, clinical traits, and medical images on a massive scale, offering millions of paired single-nucleotide polymorphisms (SNPs), thousands of clinical records, and retinal fundus photographs across hundreds of thousands of individuals–providing a unique sandbox for truly multimodal precision-medicine models (Szustakowski et al., 2021; DeBoever et al., 2020).

We introduce **PM1**[1], a multimodal foundation model trained on complete biobank-scale data that fuses genomic, clinical, and retinal imaging signals. PM1 consists of a three-stream architecture with modality-specific encoders and generative decoders, fused by a transformer encoder (Vaswani et al., 2017). The model is trained with masked language modeling (Devlin et al., 2019) and an information noise-contrastive estimation objective (InfoNCE) (Oord et al., 2018) to encourage fusion of the modality embeddings in a shared latent space for inter- and intra-modality prediction. Our training scheme allows using *all* available data despite modality-missingness, which is relevant in clinical settings, yielding up to $16\times$ more usable samples than required in complete tri-modal coverage.

We show that PM1 consistently outperforms unimodal and multimodal baselines, and adding modalities generally boosts phenotype prediction for clinical traits, as well as across the 22 autosomes. Across phenotypes, PM1 outperforms the medical foundation model MedGemma (Sellergren et al., 2025) by an average of 51% relative gain in ROC-AUC, and surpasses ContIG (Taleb et al., 2022) by an average of 19% relative gain on the full test set.

Finally, we demonstrate the usefulness of PM1 for downstream tasks. Conditioning on PM1's embeddings, synthetic retinal images can be generated, and a model trained only on generated images scores ROC-AUC **0.850** in diabetic retinopathy classification on real held-out data. In a group-level genome-wide association study (GWAS) analysis based on PM1, we are able to recover HERC2 pigmentation variants, and we achieve $>\mathbf{99}\%$ accuracy in an ancestry-inference probe, showcasing PM1's ability to capture biological structure in the fused latent space.

## 2 RELATED WORK

### 2.1 REPRESENTATION LEARNING OF BIOMEDICAL DATA

**Genotype** Recent studies in representation learning of genomic data demonstrate that deep models can capture structure in large-scale SNP data beyond linear approaches (Zhou et al., 2024a; Vivek et al., 2023; Ausmees & Nettelblad, 2022; Geleta et al., 2023), potentially uncovering complex associations, e.g., epistasis, missed by linear techniques such as PCA and penalized regression. Variational autoencoders (VAEs) (Kingma & Welling, 2013; Rezende et al., 2014), a family of deep generative models, are particularly popular for this application. VAEs estimate the mostly intractable posterior probability $p(z|x)$ over latent codes $z$ given an observed data point $x$ by maximizing the evidence lower bound (ELBO), thereby learning a factorized latent representation of the generative factors in the data. In particular, the KL divergence (Kingma & Welling, 2013; Rezende et al., 2014) in the ELBO loss enforces continuity and smoothness in the latent space—properties that are especially valuable in noisy and fully unsupervised settings such as biomedical data.

**Phenotype** Large-scale clinical datasets present unique challenges due to heterogeneous feature types (binary, categorical, continuous), systematically missing, sparsity, and measurement noise. Recent work tackles these issues with deep generative models and structure-aware embeddings. AutoComplete (An et al., 2023) trains a copy-mask denoising auto-encoder that jointly reconstructs bi-

---

[1]We publish the source code of PM1 at `https://github.com/anonymized-for-submission` (Fig. 1) to ensure reproducibility and enable follow-up work.

nary and quantitative traits, outperforming matrix-factorization baselines such as SoftImpute (Hastie et al., 2015) as well as earlier deep models HI-VAE (Nazabal et al., 2020) and GAIN (Yoon et al., 2018). POPDx (Yang et al., 2023) blends disease-ontology structure with BERT-derived text embeddings, and uses a bilinear network to assign multilabel phenotypes to individuals, retaining accuracy for rare or even unseen codes. Earlier generative approaches such as HI-VAE and GAIN remain competitive on small cohorts but scale poorly to biobank dimensions. PM1 adopts an AutoComplete-style encoder that is trained jointly with genomic and imaging modalities.

**Image** Deep learning has rapidly become the dominant paradigm for medical-image analysis, enabling fully automated disease detection, segmentation and even biomarker discovery across every major radiologic modality. Recent surveys chart this trajectory, noting the transition from early CNN classifiers to self-supervised Vision Transformers and masked-autoencoder pre-training that now routinely outperform task-specific networks while requiring far fewer labeled images (Bahr et al., 2024; Suganyadevi et al., 2021). In ophthalmology, central to our work—systematic reviews show that modern fundus and OCT models not only diagnose ocular disorders such as diabetic retinopathy or AMD, but can also predict systemic phenotypes (e.g., cardiovascular risk factors) from retinal pixels alone, underscoring the retina's value as a "window" into whole-body health (Zhou et al., 2023).

### 2.2 FOUNDATION MODELS IN BIOMEDICINE

In parallel to advances in biobanking, there is growing interest in generalist biomedical AI systems that can interpret multiple data modalities. Recent large-scale models have pushed the state of the art in medical NLP and vision-and-language tasks. Med-PaLM (Singhal et al., 2023) and Med-PaLM2 (Singhal et al., 2025), for example, are domain-specialized large language model (LLM) derived from PaLM LLM (Chowdhery et al., 2022) and fine-tuned for medical QA tasks (Truhn et al., 2024). Vision–language models such as LLaVA-Med (Li et al., 2023) and BiomedGPT (Zhang et al., 2023) extend this capability to medical imaging using multimodal transformers and instruction tuning. Med-PaLM M (Multimodal) (Tu et al., 2024) introduces an early prototype of a multimodal model combining text, image, and genomics inputs, though its use of genomic data is limited to variant calling. MedGemma (Sellergren et al., 2025) is a Gemma-3 (Team et al., 2025) checkpoint that underwent additional pretraining on medical corpora (including retinal and dermatology images, histopathology, and radiology slices), offering open-source evaluation for clinical vision-and-language tasks. Other efforts, such as ContIG (Taleb et al., 2022), explore joint image-genetic embeddings via contrastive learning.

Despite these advances, current generalist biomedical AI models still have important gaps. Vision–language models like LLaVA-Med and BiomedGPT excel at image and text understanding, but they do not incorporate structured clinical data or genomic sequences. On the other hand, text-only LLMs like Med-PaLM 2 have encyclopedic medical knowledge and reasoning ability (Truhn et al., 2024), yet remain blind to non-text modalities. Even Med-PaLM M, is a preliminary research effort whose handling of genomics addresses only a narrow task. Unlike prior vision–language models or contrastive image–genetics models, PM1 directly ingests SNP-level arrays, fuses them with phenotypes and retinal images, and supports cross-modal synthesis via modality-specific decoders, while training to handle missing modalities at the token level.

## 3 DATA

Our primary source of data is the UK Biobank (Sudlow et al., 2015), which provides rich genomic, phenotypic, and imaging records. It is a study of 500,000 adults aged 40-69 recruited across the UK in 2006-2010, providing biospecimens for genome sequencing (Halldorsson et al., 2022; Backman et al., 2021), who consented to longitudinal medical record linkage, and underwent extensive baseline phenotyping including surveys, physical measurements, and biomarker assays. Uniquely, the UKB also collected multimodal imaging on tens of thousands of participants, ranging from retinal fundus photographs to brain MRI (Elliott et al., 2018; Gulshan et al., 2016). These resources enable learning joint representations across modalities, which is the focus of PM1.

Specifically, we use unphased genomes of 438,668 UK participants with 658,720 SNP variants from the 22 autosomal chromosomes, and 3,421 tabular phenotypes encompassing clinical and lifestyle

traits, as well as several image modalities. In particular, we use $211,416$ Color Fundus Photography (CFP) images from the UKB Eye Imaging Study (Sudlow et al., 2015; Littlejohns et al., 2020; Keane et al., 2016), which provides comprehensive data for $69,600$ samples, including CFP images paired with SNP sequences and extensive phenotypic information, including various retinal conditions such as diabetic retinopathy, age-related macular degeneration, glaucoma, and cataract. To balance the three modalities in our dataset, we also incorporate samples from the EyePACS repository (Gulshan et al., 2016), which, while lacking genetic sequences and comprehensive phenotypes, contains 44,351 samples with images of their retinas and annotations for diabetic retinopathy.

# 4 PM1 ARCHITECTURE AND TRAINING SCHEME

PM1 is trained in a two-stage scheme that separates within-modality representation learning from cross-modality fusion. The first stage focuses on modality-specific encoders and decoders, while the second stage leverages these frozen representations to train a fusion model that enables cross-modality prediction via masked token modeling. A schematic overview of the architecture is provided in Fig. 1.

## 4.1 MODALITY ENCODERS AND DECODERS

Multimodal fusion of genomic, clinical, and imaging data requires robust learned representations for each modality. Let each $j$-th sample be characterized by a triplet of modalities $x_j = (x_j^{(G)}, x_j^{(P)}, x_j^{(I)})$, corresponding to genotype, phenotype, and image data. For each modality input $x_j^{(m)} \in \mathcal{X}_m$, we define an encoder-decoder pair $(\mathcal{E}_m, \mathcal{D}_m)$, where $\mathcal{E}_m : \mathcal{X}_m \to \mathbb{R}^{T_m \times d_m}$ maps raw modality-specific data to a sequence of latent $T_m$ tokens of dimensionality $d_m$, and $\mathcal{D}_m : \mathbb{R}^{T_m \times d_m} \to \mathcal{X}_m$ reconstructs the original input in the corresponding modality space. Once trained, the parameters of $\mathcal{E}_m$ and $\mathcal{D}_m$ are frozen, and only the encoder forward passes $\mathcal{E}_m(x_j^{(m)}) = z_j^{(m)}$ are used in fusion.

**Genotype encoder and decoder** $(\mathcal{E}_G, \mathcal{D}_G)$ Each sample's genotype can be represented by 22 autosomal SNP sequences $x_j^{(G)} = [x_j^{(G_1)}, ..., x_j^{(G_{22})}]$ where $x_j^{(G_c)} \in \{0, 1, 2\}^{S_c}$ denotes the SNP sequence of length $S_c$ for chromosome $c$. Given that recombination breaks linkage between chromosomes making genetic variants on different chromosomes largely uncorrelated we decide to train chromosome-specific VAEs. Each encoder $\mathcal{E}_{G_c}$ maps $x_j^{(G_c)}$ to a latent vector $z^{(G_c)} \in \mathbb{R}^{d_G}$, yielding the full genotype embedding $z_j^{(G)} = [z_j^{(G_1)}, ..., z_j^{(G_{22})}] \in \mathbb{R}^{22 \times d_G}$, after concatenation of all 22 chromosome-level embeddings. Each encoder-decoder pair $(\mathcal{E}_G, \mathcal{D}_G)$ shares the same architecture. SNPs are embedded via learned token and positional embeddings (added element-wise) into $\mathbb{R}^{S_c \times d_{\text{model}}}$. The encoder comprises three stacked residual blocks He et al. (2016) interleaved with downsampling layers; the decoder mirrors this structure with upsampling layers. Inspired by attention-free Transformer alternatives (Yu et al., 2022), each residual block contains a token mixer (depthwise convolution) to capture local LD (Flagel et al., 2019; Sheehan & Song, 2016), followed by channel mixing with feed-forward multilayer perceptrons (MLPs).

To reconstruct SNP logits, the decoder output is linearly projected back to SNP space via a learned un-embedding matrix. We optimize a weighted cross-entropy loss that accounts for allelic imbalance, with an additional KL divergence term scaled by a hyperparameter $\beta$ following $\beta$-VAE (Higgins et al., 2016). Training employs a cyclic learning rate schedule (Smith, 2017) to speed up convergence; gradient clipping to prevent exploding gradients, and KL divergence thresholding for stability (Child, 2020), along with mixed-precision and early stopping. Larger chromosomes (1–10) are trained on a single A100 GPU, while smaller ones are parallelized across up to four NVIDIA A5500 GPUs using data parallelism to increase the effective batch size. A full schematic of the architecture is provided in **Supplementary Fig. 12**.

**Phenotype encoder and decoder** $(\mathcal{E}_P, \mathcal{D}_P)$ Phenotypic measurements are collected as a single dense vector $x_j^{(P)} \in \mathbb{R}^{3,421}$, encoding a fixed list of 3,421 phenotypic binary and continuous traits. Continuous traits are standardized to zero mean and unit variance, binary traits are encoded as 0 (controls) and 1 (cases), and missing entries are imputed to zero and their location is saved for use in

the training process. The encoder $\mathcal{E}_P$ maps $x_j^{(P)}$ to a latent token $z_j^{(P)} \in \mathbb{R}^{1 \times d_P}$, and the decoder $\mathcal{D}_P$ reconstructs $p$ from this embedding. The architecture of the phenotype encoder-decoder is inspired by AutoComplete (An et al., 2023), using feedforward layers with LeakyReLU activations. The model imputes missing values and reconstructs non-missing entries. During the training process, copy masking is used to propagate realistic patterns of missingness observed in the clinical training data, masking 30% of the observed data. We minimize a mixed reconstruction loss that combines mean-squared error for continuous traits and binary cross-entropy for binary traits, evaluated only on entries that are observed or copy-masked:

$$\mathcal{L} = \frac{1}{N} \sum_{j=1}^{N} \sum_{k=1}^{P} \tilde{M}_{jk} \begin{cases} (\hat{x}_{jk} - x_{jk})^2, & k \in \mathcal{C}, \\ -x_{jk} \log \hat{x}_{jk} - (1 - x_{jk}) \log(1 - \hat{x}_{jk}), & k \in \mathcal{B}, \end{cases} \tag{1}$$

where $\hat{x}_j = \mathcal{D}_P\big(\mathcal{E}_P(x_j^{(P)})\big)$, $\mathcal{C}$ and $\mathcal{B}$ index continuous and binary traits, $M$ is the true-observation mask, $M^{\text{copy}} \sim \text{CopyMask}(p = 0.3)$ is the copy-mask, and $\tilde{M} = M \vee M^{\text{copy}}$. Gradients are back-propagated through $\mathcal{E}_P$ and $\mathcal{D}_P$ to update all parameters.

**Image encoder** $\mathcal{E}_I$   For retinal image encoding, we employ RETFound (Zhou et al., 2023). RET-Found is a state-of-the-art foundation model based on the masked autoencoder (MAE) architecture (He et al., 2022), pre-trained on a dataset of $\sim$1.6M unlabeled retinal images. To better align features with our data distribution, we perform a lightweight self-supervised adaptation on UKB images, which were not included in the pretraining, using the original MAE pixel-reconstruction objective. We attach low-rank adapters (LoRA) (Hu et al., 2022) to the attention projections and output projection in ViT blocks, freezing the base weights and updating only LoRA parameters. We use the adapted encoder $\mathcal{E}_I$ to map left/right fundus images $x_k^{(I)} = (x_k^{(I_L)}, x_k^{(I_R)}) \in \mathbb{R}^{2 \times H \times W \times 3}$ into fixed-length tokens $Z_j^{(I)} = [z_j^{(I_L)}, z_j^{(I_R)}] \in \mathbb{R}^{2 \times d_I}$; missing eyes are replaced by mask tokens.

The RETFound MAE decoder, due to the asymmetric nature of MAE where the decoder is often less powerful than the encoder, is primarily designed to support the encoder's pretraining objective (i.e., learning good embeddings) and is not optimized for high-fidelity image synthesis (Zhou et al., 2023; He et al., 2022). Instead, in the experiments section, we explore diffusion models (Sohl-Dickstein et al., 2015; Ho et al., 2020) as an alternative for this decoder.

## 4.2 FUSION MODEL $F$

After pretraining modality encoders, each $j$-th sample is represented by a triplet of latent token sequences:

$$z_j = \left[ z_j^{(G)} \in \mathbb{R}^{22 \times d_G}, z_j^{(P)} \in \mathbb{R}^{1 \times d_P}, z_j^{(I)} \in \mathbb{R}^{2 \times d_I} \right]$$

We apply token-wise standardization, project each token to a shared space $\mathbb{R}^{T \times h}$ via learned projections, and concatenate them to obtain a unified token stream $\tilde{z}_j \in \mathbb{R}^{T \times h}$, where $T$ is the total number of tokens. A learnable `[MASK]` token replaces masked entries based on a stochastic masking scheme (He et al., 2022), and optional Gaussian noise $\varepsilon \sim \mathcal{N}(0, \sigma^2 I)$ is residually added to regularize the representation, resulting in $\tilde{z}_j' = \tilde{z}_j + \varepsilon$. Finally, dropout is applied on individual dimensions of each token.

The fusion model $F : \mathbb{R}^{T \times h} \to \mathbb{R}^{T \times h}$ is a transformer encoder configured with multi-head self-attention, ReLU activations, and learned positional embeddings. In our design, we opt for a 32-layer architecture, with 8 attention heads, and a hidden dimension $h = 2048$. Layer normalization is applied prior to attention and feedforward operations. A `[CLS]` token is prepended to the input to enable global summarization and ensure that there is always at least one token that is attended. The model processes the token stream and outputs contextualized representations $\hat{z}_j$, which are then segmented back into modality-specific embeddings and passed through frozen decoders $\mathcal{D}_m$ to reconstruct the masked inputs $\hat{x}_j = (\hat{x}_j^{(G)}, \hat{x}_j^{(P)}, \hat{x}_j^{(I)})$.

The training objective involves a combination of masked token modeling $\mathcal{L}_{\text{mask}}$, denoising reconstruction $\mathcal{L}_{\text{recon}}$, and contrastive InfoNCE $\mathcal{L}_{\text{InfoNCE}}$ loss terms, jointly optimized to balance fine-grained token fidelity and modality-agnostic semantic alignment (Equation 3). The masked token modeling component operates similarly to the masked language modeling (MLM) objective in

BERT (Devlin et al., 2019), but generalized to arbitrary data modalities. For each sample, we randomly select a subset of tokens to be replaced with a learned `[MASK]` vector. The fusion model $F$ is then tasked with reconstructing the original masked inputs from the surrounding context, including both intra- and inter-modality information. $\mathcal{L}_{\text{mask}}$ encourages the model to learn conditional dependencies that span across heterogeneous sources, thereby enabling strong cross-modal imputation capabilities. In parallel, the reconstruction loss $\mathcal{L}_{\text{recon}}$ is computed on unmasked tokens and serves to stabilize training by enforcing consistency of the latent representations, even under mild input Gaussian noise perturbations or dropout applied on individual dimensions of each token. To encourage cross-modal coherence, the InfoNCE loss $\mathcal{L}_{\text{InfoNCE}}$ aligns pooled modality-specific tokens for the same sample while repelling embeddings from other samples. Specifically, the loss encourages high similarity between anchor average-pooled representations $\bar{z}_j^{(m)} = \texttt{POOL}(z_j^{(m)}) \in \mathbb{R}^{1 \times h}$ and $\bar{z}_j^{(m')}$ for different modalities $m \neq m'$ for the same $j$-th sample, while treating all cross-sample pairs $\bar{z}_k^{(m')}$ with $k \neq j$ as negatives. This formulation implicitly maximizes a lower bound on the mutual information $I(z_j^{(m)}, z_j^{(m')})$ between modality pairs under a shared fusion representation, as formalized by the InfoNCE criterion (Oord et al., 2018) (Equation 2):

$$\mathcal{L}_{\text{InfoNCE}}\big(\bar{z}_j^{(m)}\big) = -\log \frac{\displaystyle\sum_{m' \neq m} e^{\tau^{-1} \bar{z}_j^{(m)\top} \bar{z}_j^{(m')}}}{\displaystyle\sum_{\substack{m' \\ j \neq k}} e^{\tau^{-1} \bar{z}_j^{(m)\top} \bar{z}_k^{(m')}} + \sum_{m' \neq m} e^{\tau^{-1} \bar{z}_j^{(m)\top} \bar{z}_j^{(m')}}} \tag{2}$$

with a contrastive temperature $\tau > 0$. To summarize, for a batch of $N$ samples, the total fusion loss is formulated as (Equation 3):

$$\mathcal{L}_F = \sum_{j=1}^{N} \sum_{m} (1 - \lambda) \left( \mathcal{L}_{\text{mask}}(z_j^{(m)}, \hat{z}_j^{(m)}) + \mathcal{L}_{\text{recon}}(z_j^{(m)}, \hat{z}_j^{(m)}) \right) + \lambda \mathcal{L}_{\text{InfoNCE}}(\bar{z}_j^{(m)}) \tag{3}$$

where $\mathcal{L}_{\text{mask}}$ and $\mathcal{L}_{\text{recon}}$ is the mean-squared error and $\lambda \in [0, 1]$ is the weighting hyperparameter which controls the relative importance of the contrastive loss.

$F$ is trained for 25 epochs ($\approx$1 hour/epoch) with batch size 128 on a NVIDIA A100 GPU. We optimize with the quasi-hyperbolic variant of Adam (Ma & Yarats, 2018), with hyperparameters $\beta_1 = 0.9, \beta_2 = 0.999, \nu_1 = 0.7, \nu_2 = 1$ and a weight decay of $10^{-4}$, and employ a cosine-annealing learning-rate schedule with a 10-epoch linear warm-up (learning rate from $0.1 \times$ base to base) and a minimum learning rate floor of $10^{-6}$. Gradients are clipped to a max-norm of 10. We use $\lambda = 0.5$ and $\tau = 0.03$, which have shown better performance across our experiments. The total number of trainable parameters is $>$1,095 million with $>$3.5 TFLOPs per forward pass.

## 5 EXPERIMENTS AND EVALUATIONS

We comprehensively evaluate PM1 on several fronts–latent space exploration, genotype reconstruction and synthesis, retinal image synthesis, phenotype prediction and cross-modal inference–using the test split of UKB to validate both reconstructive fidelity and downstream predictive utility.

**Latent space exploration** We analyze both the modality-specific latent embeddings generated by $\mathcal{E}_m$ and the pooled fusion embeddings produced by the multimodal transformer $F$ using principal component analysis (PCA) and t-distributed stochastic neighbor embedding (t-SNE) (Van der Maaten & Hinton, 2008) for projection into two dimensions. Our results reveal that each modality contributes distinct and semantically meaningful structure to the joint representation space. Genotype embeddings, for instance, naturally cluster by ancestry without any supervision, reflecting population stratification effects encoded in autosomal variants. Similarly, phenotype embeddings display unsupervised separation by biological sex and body mass index (BMI) (see **Supplementary Figs. 5** and **6**).

More compelling insights arise from exploring the fused embeddings produced by the transformer $F$, which integrates signals from all modalities into a unified latent space. As shown in **Supplemen-**

**tary Figs. 7 and 8**, PCA on the pooled fusion tokens reveals cohesive clusters that blend ancestry, phenotypic traits, and imaging characteristics, confirming that the fusion model performs meaningful cross-modal alignment. Crucially, we observe that the inclusion of the contrastive InfoNCE loss dramatically improves this alignment: Fig. 2 compares latent fusion spaces trained with and without the contrastive objective, under identical architectures. Without contrastive supervision, the embeddings segregate cleanly by modality. In contrast, the contrastively trained model aligns corresponding modality embeddings for the same sample: for instance, left and right eye tokens from the same individual cluster tightly, demonstrating the model's success in learning modality-invariant representations. This property is particularly advantageous for missing data imputation and cross-modal generation, where shared semantics across input views is critical.

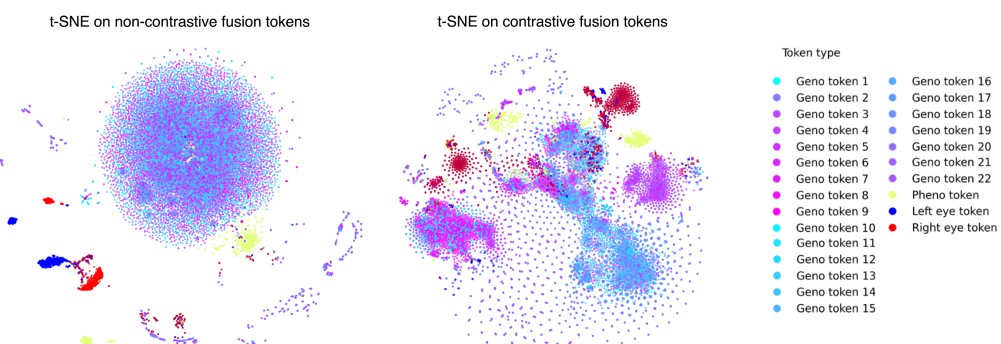

Figure 2: **t-SNE projections of token-level embeddings from the fusion model across 1,000 UK Biobank participants.** (Left) Embeddings from a model trained without the contrastive InfoNCE loss. (Right) Embeddings obtained with contrastive training. Each point corresponds to a single token–either one of 22 chromosome genotype tokens, a phenotype token, or a left and right retinal image token–colored by token type. The 2D projections illustrate clear modality-specific clustering and demonstrate the model's improved cross-modal integration when contrastive loss is applied, encouraging similar tokens across modalities to align more closely in the embedding space.

**Genotype reconstruction and synthesis**    We evaluate the capacity of our genotype encoder-decoder pair $(\mathcal{E}_G, \mathcal{D}_G)$ to reconstruct whole-genome input data and to synthesize realistic genotype samples. For reconstruction, we benchmark performance using a weighted accuracy metric that accounts for allele frequency imbalance across loci, comparing our genotype encoder architecture against two baselines: a linear projection model and a shallow VAE (**Supplementary Fig. 9**). Our genotype encoder consistently outperforms these alternatives across all autosomes.

Beyond reconstruction, we assess the realism of our synthetic genotypes by analyzing their LD structure and compare to real genotypes (Geleta et al., 2023). Specifically, we compute the folded allele frequency spectrum, showing the proportion of SNPs at each minor allele frequency (**Supplementary Fig. 9**) and compute a correlation profile based on variant distance for real and simulated sequences. Both measures suggest that our synthetic genotypes reflect patterns found in real sequences.

**Retinal image synthesis**    As an exploratory task to evaluate the richness of the learned multimodal representations, we trained a conditional diffusion model for retinal image synthesis using Denoising Diffusion Implicit Models (DDIM) (Song et al., 2022) for the backward denoising process. The core is a U-Net architecture (Ronneberger et al., 2015) inspired by successful image generation models (Rombach et al., 2022; Podell et al., 2024), composed of six downsampling and six symmetric upsampling blocks with skip connections, and output channels of 128, 128, 256, 256, 512, and 512. Input and output are 3-channels RGB images at configurable resolution. All downsampling blocks, except the second, use a stride-2 convolution (LeCun et al., 1989; Krizhevsky et al., 2012) followed by two ResNet blocks (He et al., 2016). Crucially, this U-Net incorporates cross-attention layers, which attend to keys and values derived from PM1's fusion embeddings, conditioning the image generation process on integrated multimodal representations.

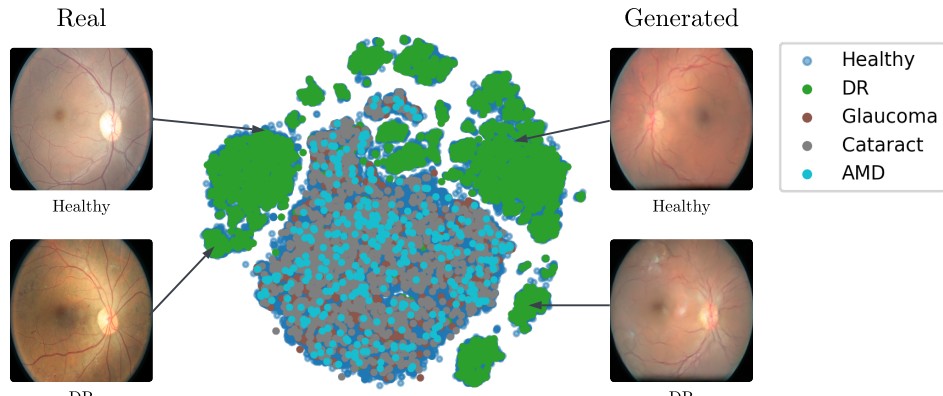

Figure 3: **Retinal image synthesis using PM1-guided conditional diffusion**. We visualize the PM1 image embeddings with t-SNE, color-coded by disease label (Healthy, Diabetic Retinopathy (DR), Glaucoma, Cataract, and Age-related Macular Degeneration (AMD)). The diffusion model is conditioned on PM1's multimodal fusion tokens via cross-attention and trained to reconstruct realistic retinal images.

Using this setup, we synthesize 50,000 retinal images (25,000 healthy; 25,000 diabetic retinopathy). A ResNet-50 trained exclusively on these synthetic images achieves ROC-AUC **0.8502** on a held-out test set of real UKB and EyePACS images. We further report Fréchet Inception Distance (FID) **50.86** and Kernel Inception Distance (KID) **0.0532**. This experiment demonstrates that the fusion model learns meaningful representations capable of guiding a complex generative task. While the scale is still exploratory and the resulting model is relatively small for producing publication-quality clinical images, the results highlight the potential of this approach if expanded with more data and computational resources.

**Cross-modal inference**   We perform cross-modal genotype inference across all 22 autosomes under different modality combinations. We confirm that incorporating more modalities consistently improves genotype prediction (for quantitative results, see **Supplementary Fig. 10**). Specifically, we test: (i) G+P+I→G, where genotype, phenotype, and image modalities are jointly provided as context to reconstruct genotype tokens; (ii) P+I→G, where genotype is inferred from phenotypic traits and retinal images; (iii) P→G and I→G, where predictive accuracy drops but still indicates that single modalities capture signals correlated with genetic variation.

For downstream trait inference, we extend evaluation to clinical phenotypes. PM1's phenotype decoder is trained to reconstruct phenotypes and predict missing ones, and can be enriched with genotype and image inputs. We compare its fusion performance in Table 1. For each target phenotype, the observed phenotypes are masked at the phenotype encoder input of PM1, as well as all other phenotype entries directly related with that trait (more details in **Supplementary Section G.1**). Phenotype prediction generally benefits from fusing the other modalities (genotype and image), boosting ROC-AUC. As additional baselines, we evaluate MedGemma (Sellergren et al., 2025) on phenotype prediction using a representative held-out subset of 1,000 UKB participants with complete P+I coverage across nine clinically relevant traits, and ContIG (Taleb et al., 2022) as an image–genetics contrastive baseline on the full test set. Table 1 shows that PM1 consistently outperforms MedGemma across all phenotypes and task settings, achieving average ROC-AUC improvements of +0.32 in the phenotype-to-phenotype (P→P) setting and +0.28 in the phenotype-plus-image-to-phenotype (P+I→P) setting. On the full test set, PM1 surpasses ContIG with an average gain of +0.12 ROC-AUC corresponding to **18.6**% relative improvement across phenotypes. More details—including the rationale for the 1,000-sample subset in MedGemma and the benchmarking setup for ContIG—are provided in **Supplementary Sections G.2 and G.3**.

### 5.1   GWAS AND GROUP-LEVEL SIGNALS IN PM1 EMBEDDINGS

We probed whether PM1's multimodal embeddings encode biologically meaningful structure. A GWAS on PCA-reduced fusion embeddings (all modalities active) using PLINK 2 (Chang et al.,

Table 1: ROC-AUC for phenotype prediction on clinical traits. Left: UKB 1,000 participants with P+I coverage comparing PM1 and MedGemma. Right: full test set with PM1 variants and ContIG.

| Phenotype | 1,000 test samples with P+I coverage | | | | Full test set | | | | |
|---|---|---|---|---|---|---|---|---|---|
| | PM1$^{P}$ | MedGemma$^{P}$ | PM1$^{P+I}$ | MedGemma$^{P+I}$ | PM1$^{P}$ | PM1$^{P+I}$ | PM1$^{P+G}$ | PM1$^{P+I+G}$ | ContIG$^{I+G}$ |
| Macular degeneration | 0.953 | 0.542 | **0.954** | 0.577 | 0.792 | **0.802** | 0.795 | 0.799 | 0.644 |
| Diabetic eye disease | **0.964** | 0.552 | 0.962 | 0.566 | 0.805 | **0.820** | 0.805 | 0.815 | 0.731 |
| Glaucoma | 0.883 | 0.473 | **0.921** | 0.604 | 0.745 | 0.750 | 0.762 | **0.766** | 0.639 |
| Cataract | 0.735 | 0.578 | **0.747** | 0.588 | 0.686 | 0.693 | 0.702 | **0.708** | 0.683 |
| Heart failure | 0.811 | 0.511 | **0.823** | 0.543 | **0.764** | **0.764** | 0.761 | 0.762 | 0.673 |
| Ischaemic stroke | 0.792 | 0.391 | **0.802** | 0.604 | 0.794 | 0.795 | 0.804 | **0.805** | 0.591 |
| Heart attack | **0.799** | 0.524 | 0.791 | 0.521 | **0.682** | **0.682** | 0.664 | 0.664 | 0.633 |
| Dementia | 0.863 | 0.583 | **0.874** | 0.590 | 0.820 | 0.819 | **0.836** | 0.834 | 0.667 |
| Alzheimer's disease | **0.845** | 0.612 | 0.844 | 0.603 | 0.818 | 0.807 | **0.828** | **0.828** | 0.668 |

2015) identified genome-wide hits in the HERC2 gene, on Chromosome 15. Variant *rs1129038* is a well-established regulatory SNP involved in human eye pigmentation. It modulates expression of the nearby OCA2 gene and is part of a founder haplotype strongly predictive of blue eye color in European populations ($p = 6.2 \times 10^{-46}$). This variant has also been linked to other ocular traits including glaucoma, macular degeneration, and central corneal thickness. A second variant, *rs1667394*, is an intronic SNP that likely influences pigmentation through OCA2 regulation and has been previously associated with adiponectin levels.

As a complementary probe, using self-reported labels, we construct a 3-class task (European, African, Asian) with a 9,152/2,288 train/test split (class-balanced by subsampling Europeans). Training simple classifiers (logistic regression, random forest, naive Bayes, k-NN) on PM1's pooled fusion embeddings achieved >**99**% test accuracy and macro-F1, confirming that PM1 captures population-level structure without ancestry supervision. See further details in **Supplementary Section C**.

## 6 CONCLUSION

We present **PM1**, the first multimodal foundation model designed for large-scale biobank data, which fuses genomic, phenotypic, and retinal imaging modalities to support cross-modal inference, generation, and representation learning. Through a range of experimental validations we demonstrate the expressiveness and versatility of the learned multimodal representations. Despite its strengths, our work bears several limitations. First, the UK Biobank cohort, while large and deeply phenotyped, is known to underrepresent global ancestry diversity, which may limit the generalizability of our model to non-European populations. Second, we highlight the inherent imbalance across modalities: images, phenotypes, and genotypes differ significantly in dimensionality, signal-to-noise ratio, and sparsity. These differences raise open questions about optimal weighting or modality-specific attention in fusion architectures. Third, the computational demands of training large transformer-based models on biobank-scale multimodal data are substantial. While we document our architecture and provide code for reproducibility, resource constraints may limit accessibility.

Nonetheless, the potential scientific and translational benefits of models like PM1 are significant. By enabling interpretable cross-modal inference and synthesis, our framework offers new avenues for modality imputation, and genotype-phenotype association discovery. In particular, this work aligns with broader goals in biomedical research, where the ability to connect rare functional genetic variation to deep phenotypic outcomes—including imaging, biomarkers, and clinical traits—can help and understand causal mechanisms (Szustakowski et al., 2021).

Finally, we recognize that models operating on high-dimensional multimodal health data carry ethical risks. Concerns around data privacy, re-identification, and disparate performance across demographic groups are especially salient in genomics and biomedical AI. As larger and more diverse datasets become available, we hope PM1 can serve as a step toward robust, equitable, and multimodally-informed precision medicine.

**Disclosure** We used large language models to aid in grammar, wording, and style improvements during the writing of this paper. The models were not used for generating ideas or analyses.

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

## A    LIMITATIONS

While PM1 demonstrates strong performance across a range of multimodal inference tasks, several limitations highlight avenues for enhancement.

**Data limitations**    First, our dataset is primarily derived from the UK Biobank (Bycroft et al., 2018) and EyePACS (Gulshan et al., 2016) cohorts, both of which significantly overrepresent individuals of European ancestry. Supplementary Fig. 4 illustrates the population distribution in the UKB, highlighting the predominance of individuals of White British ancestry, who constitute 89.2% of the cohort. As such, the generalizability of PM1 to non-European populations is limited, and performance disparities may arise when applied to ancestrally diverse cohorts. Addressing this requires future evaluation and fine-tuning on more representative datasets. Possible mitigations could also include the use of domain adaptation techniques (Muneeb et al., 2022; Comajoan Cara et al., 2024) or population-conditional resampling (Bonet et al., 2024).

Second, the input modalities—genotypes, phenotypes, and retinal images—differ markedly in terms of data density, noise characteristics, and missingness patterns. These modality-specific imbalances may lead to biased feature representations, especially when one modality dominates the signal. Moreover, the imaging modality used in PM1 is limited to retinal fundus photographs. While this choice is motivated by data availability and the known predictive power of retinal features for both ocular and systemic conditions, it restricts the usefulness of the image modality to other types of images. Importantly, our framework is modular and could be extended to other types such as X-rays or MRI, pending appropriate encoder substitution and retraining.

Supplementary Figure 4: **Population distribution of the UKB**. The cohort is heavily skewed toward individuals of European ancestry, with 89.2% of participants self-identifying as White-British.

**Architectural limitations**    Given that genetic recombination breaks linkage disequilibrium (LD) between chromosomes—rendering variants on different chromosomes largely uncorrelated—we adopt a chromosome-specific modeling strategy. Each of the 22 autosomes is encoded independently using a dedicated variational autoencoder (VAE), following the approach introduced in (Geleta et al., 2023). While this approach aligns with biological priors, it imposes limitations on modeling capacity. Compressing entire chromosomes overlooks the finer functional organization of the genome, such as gene-level or regulatory element groupings. This coarser partitioning may hinder the model's ability to uncover pathogenic associations that arise within or between biologically meaningful loci, both intra- and inter-chromosomally (Whalen et al., 2016). Moreover, due to limited compute resources, we choose to train PM1 in two sequential stages: first, modality-specific encoders and decoders are trained independently to learn within-modality representations; then, their weights are frozen before training the fusion transformer. While this design reduces computational cost, it precludes end-to-end fine-tuning across modalities and may introduce alignment mismatches. Finally, for retinal image reconstruction, we deliberately omit the RETFound decoder. This decision is moti-

vated by two key limitations: (i) the RETFound decoder is not designed for high-fidelity generation and performed poorly in our preliminary experiments, and (ii) retraining an alternative full image autoencoder is infeasible due to the proprietary nature of RETFound's pretraining data. Instead, we opt for a diffusion model conditioned on PM1's fused embeddings to synthesize images, but the available data only allows us to do an exploratory analysis of this component.

# B  SUPPLEMENTARY FIGURES

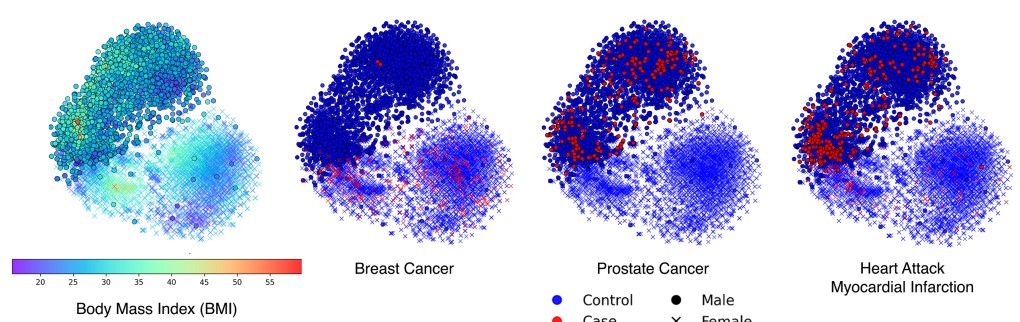

Supplementary Figure 5: **t-SNE projections of phenotype embeddings generated by $\mathcal{E}_P$ across 1,000 UKB samples**. Observe that diseases like breast and prostate cancer project onto clearly delineated axes aligned with the self-reported biological sex covariate—while breast and prostate cancers are highly correlated with respective sexes, myocardial infarction appears across sexes but with a concentration in higher BMI regions, suggesting latent structure correlating with cardiometabolic risk.

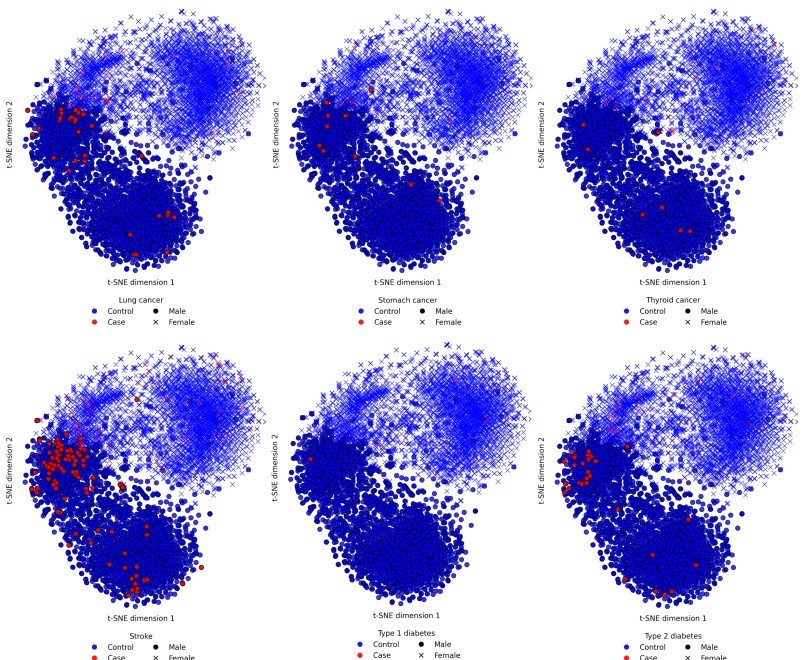

Supplementary Figure 6: **t-SNE projections of phenotype embeddings generated by $\mathcal{E}_P$ across 1,000 UKB samples, colored by positive cases of various cancers, stroke, and diabetes.** Although covariates such as sex and body mass index (BMI) are not explicitly provided as input, the learned embeddings reflect their influence through emergent structure. Distinct clusters and phenotype-specific separations indicate that the model captures latent demographic and physiological factors associated with disease risk.

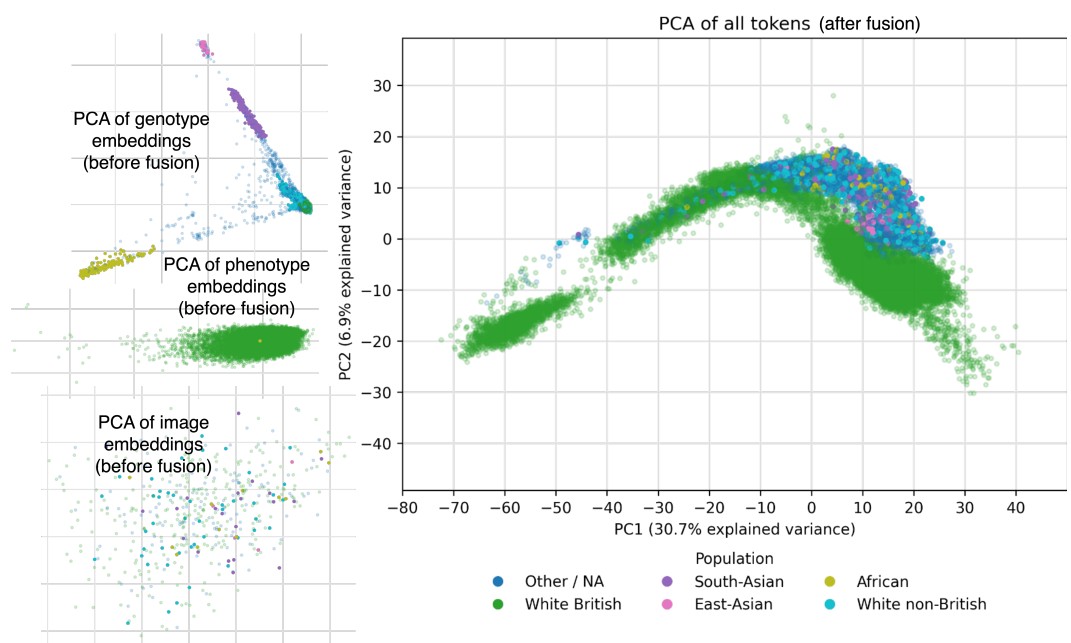

Supplementary Figure 7: **Visualization of modality-specific and fused latent representations in PM1 across >30,000 UKB samples.** The left panel shows PCA projections of pre-fusion latent embeddings, color-coded by ancestral origin. The right panel plots the post-fusion embeddings, illustrating how the transformer fusion network aligns heterogeneous modalities into a unified representation. This demonstrates PM1's capacity to integrate semantically rich but structurally diverse inputs into a coherent latent space.

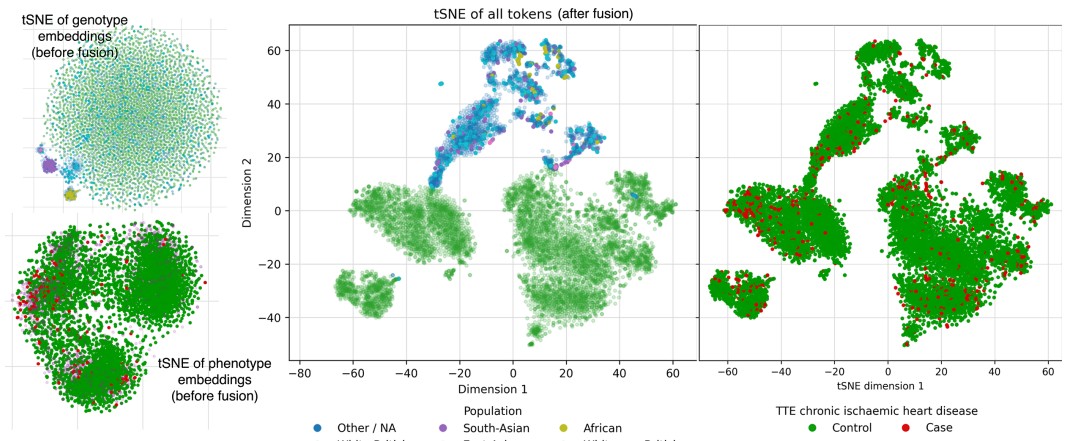

Supplementary Figure 8: **t-SNE projections of PM1's latent space before and after modality fusion across >30,000 UKB samples.** The left panel shows t-SNE projections of genotype and phenotype embeddings independently, where genotype embeddings are color-coded by ancestry, while the phenotype embeddings are color-coded by an indicator on chronic ischaemic heart disease condition. In the middle and right panels we have the t-SNE-embedded genotype/phenotype/image tokens after fusion, color-coded by ancestry and heart disease, respectively.

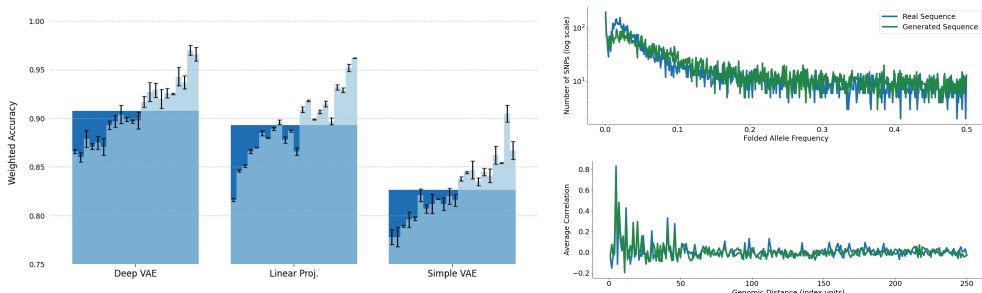

Supplementary Figure 9: **Genotype reconstruction and synthesis evaluation metrics**. (Left) Per-chromosome ordered from left to right by size (light blue) and mean reconstruction accuracy (dark blue), comparing our custom genotype encoder to a linear projection model and a shallow VAE. (Right) Assessment of the realism of synthetic genotypes generated by our genotype Encoder. We compare the folded allele frequency spectrum and the LD decay–measured via correlation with neighboring variants–between real (blue) and synthetic (green) genotypes.

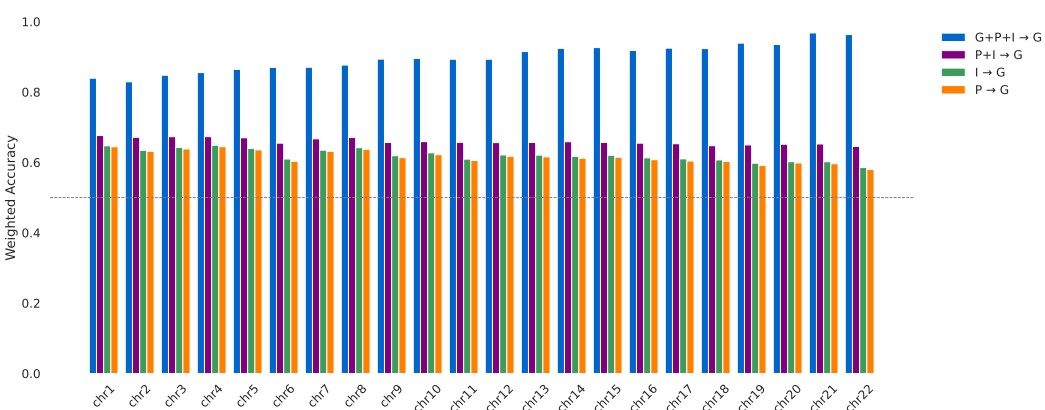

Supplementary Figure 10: **Cross-modal genotype inference across all 22 autosomes under different modality combinations**. Each group of bars corresponds to a specific chromosome (chromosome 1 through chromosome 22), and each bar within a group represents a distinct cross-modal input configuration used to reconstruct the genotype sequence for that chromosome.

## C  ANCESTRY INFERENCE AS A PROBING TASK

To demonstrate the usefulness of the learned representations, we performed an additional downstream evaluation focused on continental-ancestry inference. Starting with UKB participants' self-reported ancestry, we defined a target variable consisting of three ancestral classes, specifically: European (comprising *White British* samples), African, and Asian (including both *East Asian* and *South Asian*). To mitigate class imbalance we randomly subsampled 10,000 (from a total of 35,181) European individuals while retaining every African (587) and Asian (853) sample, yielding a cohort we split into 9,152 train and 2,288 test samples. For each individual we extracted the pooled Fusion embeddings produced by PM1 and concatenated them into a single latent representation (a 6,144-dimensional vector that integrates genotype, phenotype, and retinal information) and trained four off-the-shelf standard machine learning classifiers–logistic regression, random forest, naive Bayes, and $k$NN (with $k \in \{3, 10, 25\}$).

All models surpassed >**99**% test and balanced accuracy, macro-F1>**99**%, and weighted-F1>**99**%, confirming that ancestral groups are cleanly separable in the learned embedding space. These results demonstrate that PM1 captures population-level structure without any ancestry supervision and its representations transfer effectively to an independent classification task.

## D  ETHICAL IMPLICATIONS AND SOCIETAL IMPACTS

The development and deployment of multimodal biomedical foundation models like PM1 raises important ethical considerations that warrant careful examination. As these models integrate highly sensitive genetic, clinical, and imaging data at unprecedented scale, we must address both the transformative potential for advancing precision medicine and the risks of perpetuating or amplifying existing healthcare disparities.

### D.1  POTENTIAL RISKS

**Privacy and Re-identification Risks** PM1 encodes highly sensitive personal health data—including genome sequences, phenotypic traits, and medical images—into dense multimodal representations. Although UKB data is de-identified at the source (Bycroft et al., 2018), concerns remain regarding the risk of re-identification from model outputs or learned representations. In particular, model inversion attacks, which aim to reconstruct original training data by exploiting access to a trained model, could potentially leak sensitive information (Rigaki & Garcia, 2023; Zhou et al., 2024b). For instance, (Fredrikson et al., 2014) demonstrated that pharmacogenetic models could leak sensitive genetic markers. However, successfully reconstructing entire high-dimensional training samples, such as full genomes, from model outputs or embeddings presents substantial challenges. Recent surveys on model inversion attacks highlight that successful model inversion attacks on complex, high-dimensional data typically rely on exploiting specific model vulnerabilities or having access to substantial prior knowledge, rather than a simple decoding of learned representations (Rigaki & Garcia, 2023; Zhou et al., 2024b).

The model's cross-modal inference capabilities also raise particular concerns about genetic discrimination (Moreau, 2019). Although legislation like the Genetic Information Nondiscrimination Act (GINA) (110th United States Congress, 2008) offers some protection, PM1's ability to infer genetic information from non-genetic modalities (e.g., predicting SNPs from retinal images or phenotypes) could potentially circumvent these safeguards (Shi & Wu, 2017). Furthermore, as foundation models like PM1 compress data from numerous individuals, ensuring data privacy and compliance with regulations becomes paramount. This includes the challenge of implementing methods for selectively "forgetting" patient data upon request (Carrillo-Perez et al., 2024).

**Representational Bias and Health Disparities** PM1 is trained predominantly on UKB data (Bycroft et al., 2018), which overrepresents individuals of European ancestry. This creates a concrete risk of performance disparities: the model's predictions are likely to be less accurate for individuals from underrepresented populations (Martin et al., 2019). Since disease prevalence, presentation, and genetic architecture vary across ancestries, a model trained primarily on European data may produce biased clinical predictions for non-European individuals, potentially exacerbating existing healthcare inequities. While (Carrillo-Perez et al., 2024) suggest that synthetic data generated by such models *after* training might offer a way to balance datasets, the initial training on biased data remains a core challenge that could lead to the perpetuation of existing biases.

Furthermore, the quality and nature of the input data significantly impact model performance and the reliability of its outputs. As highlighted by (Carrillo-Perez et al., 2024) a key concern arises when data *missing not at random* (MNAR)—that is, the likelihood of data being missing is related to its actual value or other unobserved factors—models trained on such data may perpetuate biases or generate inaccurate imputations. For instance, if certain tests are more frequently performed on sicker patients, the model might learn a skewed representation of the general population, impacting the fidelity of generated or imputed data for less represented or healthier subgroups.

### D.2  POSITIVE APPLICATIONS

Despite these concerns, multimodal foundation models like PM1 offer significant potential benefits for advancing healthcare and biomedical research:

**Democratizing Access and Reducing Burden** PM1's ability to infer information across modalities can enhance accessibility to advanced medical insights. By inferring genetic risk from more readily available and less expensive data, such as retinal images or clinical records, it could reduce

reliance on costly or invasive procedures. Indeed, it has been proposed that such models could obviate the need for invasive procedures by imputing desired information from already collected non-invasive data. Moreover, they offer a way to lessen the economic strain associated with acquiring certain medical modalities. This could democratize aspects of precision medicine, especially in resource-limited settings, and make complex diagnostics more widely available (Carrillo-Perez et al., 2024).

**Enhancing Early Diagnosis and Personalized Medicine** Multimodal integration, as demonstrated by PM1, enables a more comprehensive and holistic assessment of disease risk. Combining genetic predispositions with subtle changes in imaging or clinical data could lead to earlier identification of at-risk individuals, often before clinical symptoms manifest, aligning with the goals of precision medicine (Acosta et al., 2022).

**Accelerating Research and Discovery** The integrated representations learned by PM1 can significantly accelerate biomedical research. The model's capacity for cross-modal inference might help reveal novel genotype-phenotype associations and guide hypothesis generation. PM1 has the potential to be used for "in silico hypothesis testing", (Carrillo-Perez et al., 2024), where researchers could study the effects of altering specific features in one modality on others within a simulated environment. This, coupled with improved data imputation and the generation of high-quality synthetic samples, can enhance the diversity and availability of data for research, particularly in scarce-data settings (Carrillo-Perez et al., 2024). The integration of these data is expected to substantially improve our comprehension of human health and enable the development of more precise, personalized approaches to prevention, diagnosis, and treatment (Karczewski & Snyder, 2018; Acosta et al., 2022).

PM1, as introduced in this paper, exemplifies the increasing capability of multimodal foundation models to contribute to precision medicine. The integration of diverse biomedical data sources offers clear avenues for enhanced diagnostic insight and accelerated research. However, such advancements are intrinsically linked with significant ethical considerations, including data privacy, security, and the imperative to avoid perpetuating health inequities through biased representations. It is crucial that ongoing and future research in this field embeds rigorous ethical frameworks and fosters interdisciplinary collaboration to navigate these complexities and guide the responsible development of these impactful technologies.

# E DATA PROCESSING

## E.1 DATA SPLIT

We adopt a fixed 80/10/10 split of our 521,269 samples for training, validation, and testing, using a random seed of 42. This split is applied to the outer join of all available samples across modalities, ensuring consistent partitioning for the fusion model. For training the modality-specific encoders, we use the same global split and select the corresponding 80%, 10%, and 10% subsets of each individual modality that fall into the respective buckets. This strategy ensures alignment between modality-specific and fusion training while preventing data leakage across evaluation stages. Supplementary Tables 2, 3, and 4 show summary statistics of genotype (G), image (I), and phenotype (P) data availability across training, validation, and test splits, including sample counts and missingness for each disease phenotype.

## E.2 GENOTYPE DATA

For the genotype data, we use UKB SNP genotype arrays spanning 487,409 individuals, each represented with 658,720 variants. The SNP arrays in the UK Biobank are significantly shorter than the total number of known variants (Halldorsson et al., 2022), as not all positions were initially sequenced. For this initial version of the foundation model, we decide to rely on the shorter SNP arrays, which are more widely used and readily accessible.

Supplementary Table 2: Modality missingness per data split.

| Split | $n_{samples}$ (% of total) | $n_{geno}$ (missing) | $n_{img}$* (missing left, missing right) | $n_{pheno}$ (missing**) |
|---|---|---|---|---|
| Train | 417,015 (80%) | 350,753 (66262) | 91,212 (left: 333,124, right: 331,585) | 269,617 (147,398) |
| Val | 52,127 (10%) | 44,055 (8072) | 11,354 (left: 41,709, right: 41,524) | 33,771 (18,356) |
| Test | 52,127 (10%) | 43,860 (8267) | 11,385 (left: 41,676, right: 41,504) | 33,741 (18,386) |
| SUM | 521,269 (100%) | | | |

\* Number of samples for which at least one retina scan is available.
\*\* Number of samples for which no phenotype data is available. Phenotype-specific missingness can be found in the phenotype rows.

Supplementary Table 3: Phenotype (eye diseases) counts per split shown as case (positive), total (case and control), and missing.

| Split | Macular degeneration | | | Diabetic eye | | | Glaucoma | | | Cataract | | |
|---|---|---|---|---|---|---|---|---|---|---|---|---|
| | Case | Total | Miss | Case | Total | Miss | Case | Total | Miss | Case | Total | Miss |
| train | 433 | 380,857 | 7 | 1,021 | 380,269 | 7 | 6,256 | 126,401 | 248,640 | 16,640 | 125,092 | 239,565 |
| val | 50 | 47,753 | 1 | 141 | 47,662 | 1 | 782 | 15,780 | 31,242 | 1,996 | 15,624 | 30,184 |
| test | 49 | 47,669 | 0 | 132 | 47,586 | 0 | 769 | 15,754 | 31,195 | 2,053 | 15,594 | 30,071 |

Supplementary Table 4: Phenotype (cardiovascular diseases and neurodegenerative diseases) counts per split shown as case (positive), total (case and control), and missing.

| Split | Heart failure | | | Ischaemic stroke | | | Heart attack | | | Dementia | | | Alzheimer's disease | | |
|---|---|---|---|---|---|---|---|---|---|---|---|---|---|---|---|
| | Case | Total | Miss | Case | Total | Miss | Case | Total | Miss | Case | Total | Miss | Case | Total | Miss |
| train | 7,082 | 37,4208 | 7 | 4,557 | 376,734 | 6 | 15,802 | 365,488 | 7 | 436 | 380,855 | 6 | 745 | 380,546 | 6 |
| val | 898 | 46,905 | 1 | 613 | 47,190 | 1 | 1,966 | 45,837 | 1 | 57 | 47,746 | 1 | 101 | 47,702 | 1 |
| test | 903 | 46,815 | 0 | 584 | 47,134 | 0 | 2,017 | 45,701 | 0 | 61 | 47,657 | 0 | 89 | 47,629 | 0 |

Genetic data is split by chromosome and processed using the PLINK 2 format (Chang et al., 2015). Each SNP $s$ is represented in a ternary system $s \in \{0, 1, 2\}$ and typically stored in 32-bit integers. To reduce memory and computation overhead, we compress every 4 SNPs into a single 8-bit integer—effectively using 2 bits per SNP—which results in a 16-fold memory reduction. This reduces the RAM requirements to use the data and accelerates both data processing and GPU transfer.

### E.3 PHENOTYPE DATA

We process a wide variety of traits recorded in the UK Biobank ranging from disease outcomes, cancer registry data, family history, biomarkers, and other binary and quantitative measurements, following a similar procedure as in (Tanigawa et al., 2022). Age, sex, and the first 10 principal components (PCs) of genotypes are included as columns. All phenotype columns are further processed to obtain quantitative and binary columns, allowing missing values. Phenotype information that is registered as categorical with more than two categories is one-hot encoded and treated as binary. Quantitative traits are $z$-score normalized with statistics computed on the training set, and phenotype columns with $\leq 1$ non-missing unique value in the training data are removed. After preprocessing, we retained only the set of features present in both the training and validation splits and fixed a common column order for all splits.

### E.4 IMAGE DATA

The image preprocessing sequence involves multiple stages to isolate and standardize the retinal region. Initially, each image is loaded and converted to a grayscale representation. To mitigate noise, a Gaussian blur is subsequently applied. This blurred grayscale image is then binarized using a thresholding technique: pixel intensity values exceeding 20 (on a 0-255 scale) are set to white, while all other pixels are set to black, thereby creating a binary mask. Contours are identified within this mask utilizing the `findContours` function from the OpenCV library (Bradski, 2000). The

contour encompassing the largest area is heuristically considered to be the retina. Following the identification of the retinal contour, its bounding box is determined, and from this, the center of the retina is computed. The dimensions for cropping are then established by taking the maximum of the bounding box's width and height, and incorporating an additional 5-pixel margin on each side.

The original color image is then cropped according to these calculated dimensions, carefully ensuring that the cropped area is contained within the original image boundaries and that the retina is positioned centrally. As a final step in this phase, the cropped image is resized to a target resolution of 587×587 pixels and stored in PNG format. It is important to emphasize that the initial grayscaling, blurring, and thresholding operations are performed on a temporary version of the image solely for the purpose of accurate retina localization; the color information of the final CFP image remains unaltered throughout this process.

An incidental outcome of this preprocessing methodology was the automatic exclusion of certain images. Specifically, if no contour could be detected, the image was omitted from further processing. Manual examination of these discarded images revealed them to be entirely black, rendering their exclusion advantageous for the subsequent analysis. Given the extensive size of the dataset and the author's limited domain expertise in retinal pathology, no additional manual curation or filtering of the images was undertaken.

Subsequent to these initial preprocessing steps, and prior to their input into the machine learning models, the images undergo further transformations facilitated by the torchvision library (Ansel et al., 2024). These transformations consist of resizing the images to a more compact resolution, converting them into PyTorch tensor format, and normalizing their pixel values. This normalization, which adjusts pixel values to achieve a zero mean and unit variance, is a common practice that typically enhances the stability and convergence rate of model training.

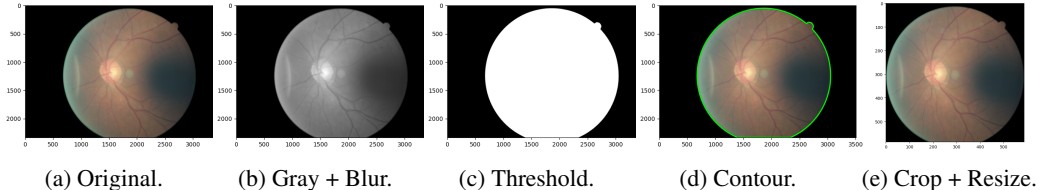

| (a) Original. | (b) Gray + Blur. | (c) Threshold. | (d) Contour. | (e) Crop + Resize. |

Supplementary Figure 11: **Demonstration of the image preprocessing pipeline**. The pipeline is applied to the CFP image of the left retina of sample 498 in the EyePACS dataset (Dugas et al., 2015; Gulshan et al., 2016).

### E.5 FUSION OF MODALITIES

The multimodal dataset consists of $N_G = 487,409$ genotype samples, $N_P = 476,569$ phenotype samples, and $N_I = 211,416$ retina image samples, of which 97,465 include both eyes and 113,951 include only one. To maximize data utilization, we perform an outer join across all modalities instead of an inner join, which would restrict the dataset to only 47,432 samples, to align the data. Naturally, this implies that many samples will have missing modalities. To allow the fusion model to handle such missing data, we pre-compute a binary data mask $M_{\text{data}} \in \{0,1\}^T$ for each sample, indicating the presence (1) or absence (0) of each sequence position. Missing tokens are replaced with a learned mask embedding. In addition, we define a loss mask $M_{\text{loss}} \in \{0,1\}^T$ for each sample, which identifies the token positions used in the masked modeling objective. For each sample, the loss mask $M_{\text{loss}} \in \{0,1\}^{25}$ is sampled from a Bernoulli distribution with success probability $p_{\text{mask},\text{mod}_i,\text{seq}_j}$, independently of all sequence positions.

Let $n_{\text{mod}} = 3$ denote the number of modalities: genotype ($G$), phenotype ($P$), and image ($I$). Each modality $\text{mod}_i \in \{G, P, I\}$ is masked with equal probability (Supplementary Equation 4):

$$p_{\text{mask},\text{mod}_i} = \frac{1}{n_{\text{mod}}} = \frac{1}{3} \tag{4}$$

Each modality contains a different number of tokens: $n_{\text{seq},G} = 22$, $n_{\text{seq},P} = 1$, and $n_{\text{seq},I} = 2$. To evenly distribute the modality masking probability across its tokens, we define the per-token masking probability as (Supplementary Equation 5):

$$p_{\text{mask},\text{mod}_i,\text{seq}_j} = p_{\text{mask},\text{mod}_i} \cdot \frac{1}{n_{\text{seq},\text{mod}_i}} \tag{5}$$

This results in per-token masking probabilities of approximately 0.015 for each genotype sequence position, 0.333 for the phenotype sequence position, and 0.167 for each image sequence position. The loss mask is resampled independently for each sample in every epoch.

Before feeding the modality-specific embeddings into the fusion model—or reconstructing them using the respective decoders—we standardize each modality to have zero mean and unit variance to mitigate the impact of scale mismatches between modalities, which can lead to imbalanced gradient contributions during loss optimization. This normalization is performed using the mean and standard deviation computed on the training set only, ensuring no information leakage into the validation or test sets.

# F    MODEL SPECIFICATIONS

This section provides a detailed overview of the model architecture, training protocol, and hyper-parameter choices for PM1's modality-specific encoders and decoders, and the transformer-based fusion network, as used throughout our experiments. We emphasize transparency and reproducibility by documenting our design rationale and training infrastructure.

## F.1    GENOTYPE ENCODER

We design a modality-specific encoder-decoder architecture for genotype data, implemented as a VAE. A schematic overview of the model architecture is provided in Supplementary Fig. 12. The architecture leverages a compact and expressive latent space of dimensionality `d_latent` = 2048, optimized to capture high- and low-level genomic features while minimizing information loss. Each SNP position is embedded into a $d_{\text{model}} = 64$-dimensional vector using the sum of learned token and positional embeddings (Vaswani et al., 2017). For a chromosome with $S_c$ SNPs, the embedded input lies in $\mathbb{R}^{S_c \times 64}$.

The encoder $\mathcal{E}_G$ and decoder $\mathcal{D}_G$ consist of `num_blocks` = 3 blocks, each composed of `block_depth` = 3 residual sub-layers. These blocks are separated by spatial resolution-changing layers (downsampling in the encoder and upsampling in the decoder) to construct a hierarchical representation, reflecting the hierarchical organization of genomic data (Crick, 1970). The number of channels changes by a factor of `seq_change_rate` = 2 at each resolution change. To ensure compatibility with the hierarchical structure of the model, the input sequence of each chromosome is padded to the next larger number divisible by `seq_change_rate`$^{\text{num\_blocks}}$ $= 2^3 = 8$. The reconstructed output for the padded sequence positions is discarded for evaluation.

Inspired by recent attention-free Transformer alternatives (Yu et al., 2022; Zhai et al., 2021), each residual sub-layer is composed of two modules: a token-mixer and a channel-mixer. The token-mixer is implemented as a sequence of four depthwise convolutional layers. The first and last layers preserve the input channel dimension, while the two intermediate layers expand the channel size by a factor of `conv_mult` = 1.5 before reducing it back, enhancing local receptive fields as seen in RNNs (Bai et al., 2018). This structure is particularly well-suited for capturing local dependencies in genomic sequences such as those introduced by linkage disequilibrium (LD). The channel-mixer is a three-layer feed-forward MLP (Rumelhart et al., 1986), where the hidden layer width is increased by a factor of `mlp_mult` = 2.0 relative to the input and output dimensions. Residual connections (He et al., 2016) are included in both mixers to stabilize training, while normalization layers such as batch norm (Ioffe & Szegedy, 2015) are intentionally excluded due to observed instabilities. The output of the final encoder block is flattened and passed through a linear projection layer that maps it to the parameters of a Gaussian distribution: a mean vector $\boldsymbol{\mu}$ and a log-variance vector $\log \boldsymbol{\sigma}$, each of dimension `d_latent` = 2048. The latent representation $\mathbf{z}$ is then sampled using the reparameterization trick (Kingma & Welling, 2013; Rezende et al., 2014), enabling backpropagation

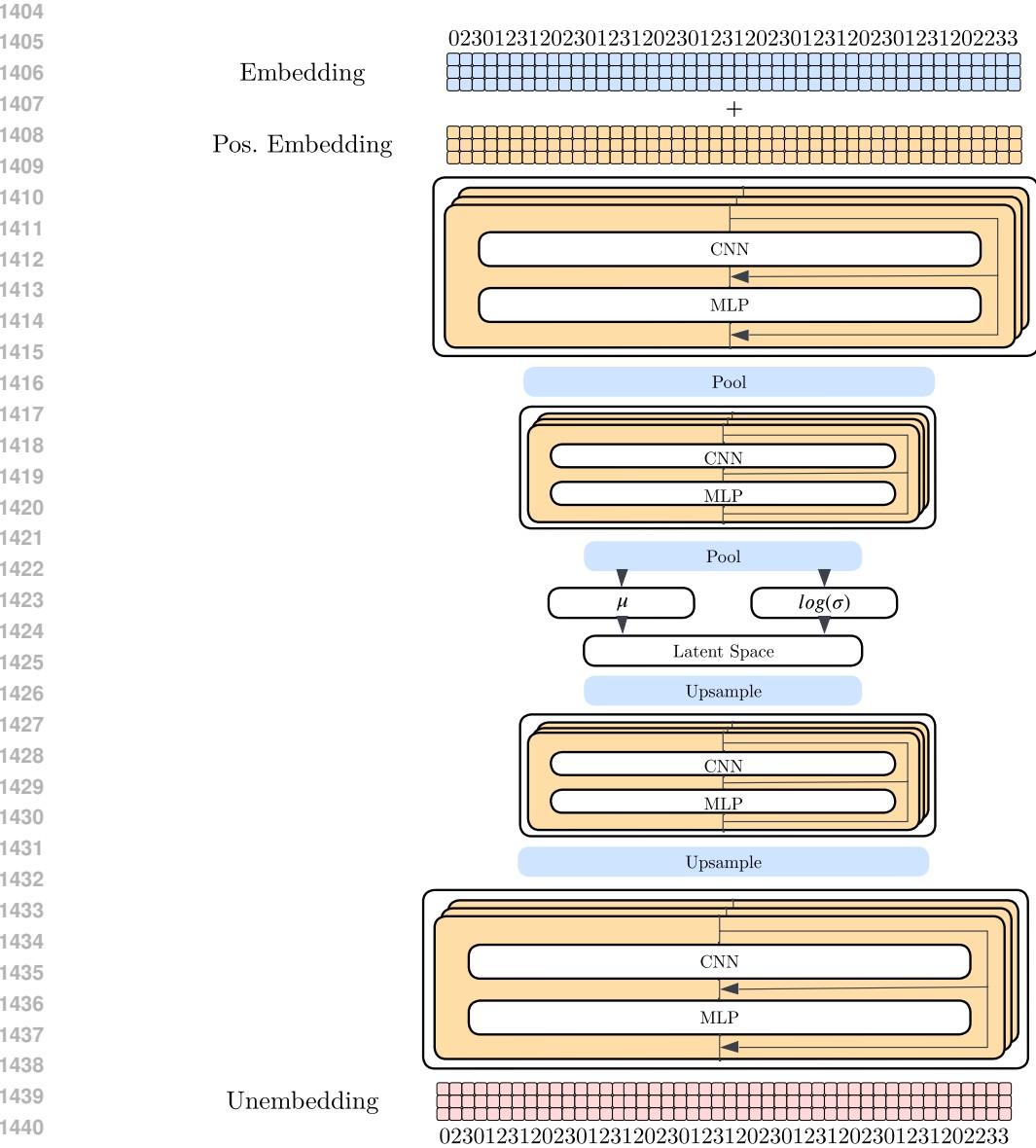

Supplementary Figure 12: **Schematic overview of genotype VAE architecture.**

through the stochastic sampling step. The reconstructed output, after running through the decoder $\mathcal{D}_G$, is projected back into the original SNP input space using a learned un-embedding matrix, producing logits for each SNP.

The training loss used to train the genotype encoder entails a weighted cross-entropy term to correct for allelic imbalance and a KL divergence regularizer scaled by `beta_kl` $= 0.0913$, in accordance with the $\beta$-VAE formulation (Higgins et al., 2016). KL divergence is annealed over the first `kl_anneal_steps` $= 1000$ steps to prevent divergence of the KL divergence loss within the first few training steps.

The training process uses a cyclic learning rate schedule (`CyclicLR`) with a triangular mode. The learning rate cycles between `base_lr` $= 1.21 \times 10^{-6}$ and `max_lr` $= 3 \times 10^{-6}$ with a step size of `step_size` $= 2000$ updates. As the optimizer we use Adam (Ma & Yarats, 2018) with the default hyperparameters `betas` $= (0.5, 0.999)$. Mixed-precision training is enabled using the `float16` datatype to allow for larger batch sizes (`batch_size` $= 32$) on available hardware.

To avoid instability during training, gradient clipping is applied with a maximum norm of `max_norm = 1.0` and steps in which the KL divergence loss includes NaN values are skipped. Training runs for up to `epochs = 20` epochs, with early stopping applied if no validation improvement is observed for `patience = 5` epochs.

Training is conducted on a single NVIDIA A100 GPU for large chromosomes (1–10), and parallelized across up to four NVIDIA A5500 GPUs using data parallelism for smaller chromosomes. Training times range from 13.22 hours (chromosome 1, 13 epochs) to 6.84 hours (chromosome 22, 20 epochs), depending on sequence length and convergence behavior.

### F.2 PHENOTYPE ENCODER

We instantiate $(\mathcal{E}_P, \mathcal{D}_P)$ as a denoising auto-encoder with 5 fully-connected layers on the encoder side and a symmetric decoder. Layer widths decrease linearly from $3{,}421$ inputs to a $2{,}048$-dimensional bottleneck (`depth=5`, `latent_dim=2048`). Each linear layer is followed by a LeakyReLU activation ($\alpha = 0.01$). Model parameters are optimized with stochastic gradient descent (SGD); gradient norms are clipped to 10. Training runs on a single NVIDIA A5500 (24 GB) GPU with early stopping after 50 epochs without improvement in validation loss, which corresponds to roughly 24-hour wall-clock time. During training, every input row is corrupted with copy-masking with probability 0.3, where the observed/missing pattern of a randomly chosen donor row from the training set is applied and the newly masked entries are set to zero. During evaluation for a given target phenotype, the column corresponding to that phenotype and all other phenotypes that are directly related are fully masked.

Hyperparameters were selected via Hyperband (Li et al., 2018) with up to 150 trials minimizing validation loss. The search space comprised: learning rate $\in [10^{-3}, 10^{-1}]$ (log-uniform), batch size $\in \{512, 1024, 2048\}$, depth $\in \{4, \ldots, 9\}$, and momentum $\in \{0.8, 0.9, 0.95\}$. The best configuration (learning rate $= 4.78 \times 10^{-2}$, batch size $= 512$, depth $= 5$, and momentum $= 0.8$, was adopted for all results reported in the paper.

### F.3 IMAGE ENCODER

PM1's retinal image encoder is adapted from the pre-trained RETFound model (Zhou et al., 2023). This encoder is a Vision Transformer (ViT) (Dosovitskiy et al., 2020), specifically configured to process inputs at a resolution of 224x224 pixels, utilizing a patch size of 16x16 pixels. Architecturally, it comprises 24 Transformer blocks, each with an embedding dimension of 1024. The pre-training regimen for this encoder involved an initial phase on natural images sourced from the ImageNet-1k dataset (Deng et al., 2009). This was subsequently followed by continued training on an extensive collection of 904,170 CFP retinal images via masked image modeling (He et al., 2022). The majority of these retinal images were drawn from the proprietary Moorfields Diabetic Image Dataset (MEH-MIDAS), as detailed in the RETFound publication (Zhou et al., 2023). The specific pre-trained checkpoint leveraged in PM1 was acquired from the HuggingFace Hub, corresponding to the repository ID `YukunZhou/RETFound_mae_natureCFP`.

We adapt the RETFound image encoder (checkpoint `RETFound_mae_natureCFP.pth`) to UKB fundus images using by inserting LoRA adapters (Hu et al., 2022) (rank $r$=8, $\alpha$=16, dropout 0.1) into the self-attention projections (`qkv`) and the output projection (`proj`) in all ViT blocks with the PEFT (Mangrulkar et al., 2023) library. Base weights remain frozen; only LoRA parameters are trainable. We train for 10 epochs on the UKB training split (inputs $224 \times 224$, ImageNet normalization), using AdamW (learning rate $1 \times 10^{-4}$) and batch size 256 with the original masked-autoencoding reconstruction loss with mask ratio 0.75 and without any labels. After training, we merge the adapters into the encoder weights. The adapted encoder produces left/right tokens $z^{(I_L)}$, $z^{(I_R)}$ consumed by the fusion transformer. The MAE decoder is not used within PM1; diffusion models handle image generation.

### F.4 DIFFUSION MODEL

The diffusion model employed for retinal image synthesis, as explored in Section 5, implements the Denoising Diffusion Implicit Models (DDIM) scheduler (Song et al., 2022) for the diffusion process, incorporating the specific modifications detailed in (Lin et al., 2024), and the denoising

process is based on a conditional U-Net architecture (Ronneberger et al., 2015), drawing inspiration of previous works (Rombach et al., 2022). We utilize 1000 diffusion steps during training and 50 denoising steps during inference.

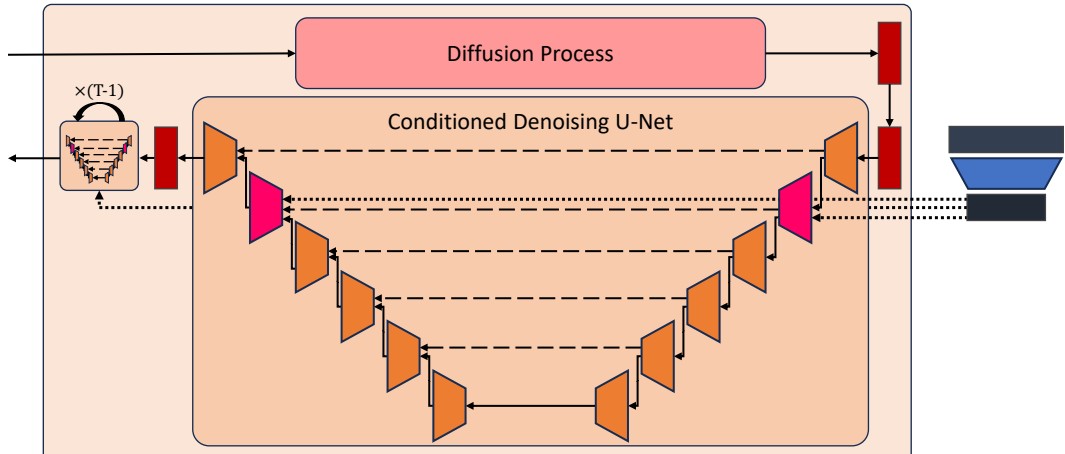

Supplementary Figure 13: **Overview diagram of the diffusion component architecture**. The model is composed of a forward diffusion process (DDIM (Song et al., 2021)), and a conditioned denoising U-Net (Ronneberger et al., 2015) which is applied for $T$ timesteps to reverse the diffusion process. The trapezoids shrinking from right to left are downsampling blocks, and the ones growing are upsampling blocks. The pink ones, additionally, incorporate cross-attention (Vaswani et al., 2017) between the hidden states of the model and the embedding of the condition. The dashed lines represent skip connections, and the dotted lines the conditioning with the embedding. A linear layer (in blue on the right) is used to obtain the embedding used for conditioning.

Our implementation of the U-Net (Ronneberger et al., 2015) consists of six downsampling blocks and six symmetrically structured upsampling blocks. Skip connections are utilized to concatenate feature channels from the contracting path to the corresponding layers in the expanding path. The sequence of feature channels produced at the output of these blocks is 128, 128, 256, 256, 512, and 512, respectively (these are visually distinguished as orange or magenta in Supplementary Fig. 13). The model is designed to process 3-channel RGB 224x224 images. Except for the second downsampling block, each such block comprises a convolutional layer with a stride of 2 (to achieve spatial dimension reduction), which is then followed by two ResNet blocks (He et al., 2016) incorporating SiLU activation functions (Hendrycks & Gimpel, 2016).

To enable conditional image generation, the U-Net architecture integrates cross-attention layers (highlighted in magenta in Supplementary Fig. 13). These layers attend to key and value projections derived from the embeddings generated by PM1's fusion model. This mechanism effectively conditions the image synthesis process on the integrated multimodal representations. The fusion embeddings are projected to a dimensionality of 768, matching that of the attention keys and values, through a linear layer (indicated in blue on the right of Supplementary Fig. 13) which also utilizes a SiLU activation function (Hendrycks & Gimpel, 2016). To guide the diffusion process via these embeddings, we employ classifier-free guidance (Ho & Salimans, 2021).

The model parameters were optimized using the AdamW optimizer (Loshchilov & Hutter, 2019) for a total of 100 epochs. The learning rate was managed by a cosine annealing schedule coupled with a linear warmup phase (Loshchilov & Hutter, 2017). To ensure numerical stability and prevent issues like exploding gradients, gradient clipping with a maximum norm of 1 was applied, and gradient accumulation was performed over 10 steps. Furthermore, following practices from models such as Stable Diffusion XL (Podell et al., 2024), an Exponential Moving Average (EMA) of the model weights is maintained (Ruppert, 1988; Polyak & Juditsky, 1992). To optimize computational performance, the diffusion model's attention mechanisms leverage the FlashAttention-2 algorithm (Dao, 2024), implemented via the xFormers library (Lefaudeux et al., 2022). To further mitigate memory demands during the training of the diffusion model, mixed-precision training with `bf16` is used.

The diffusion model was trained using data parallelism distributed across 4 NVIDIA A100 GPUs, which provided a cumulative total of 320 GB of GPU memory. The decision to not pursue a larger or more complex model architecture was predicated on the available dataset size and computational resource limitations. Moreover, the primary goal of this phase was an exploratory investigation into the feasibility and potential of conditional retinal image synthesis using the described multimodal inputs, rather than an exhaustive optimization of model scale or performance.

### F.5 FUSION MODEL

The PM1 fusion model is instantiated as a transformer encoder architecture that operates over tokenized embeddings derived from genotype, phenotype, and retinal image modalities. Specifically, we use a 32-layer transformer with a hidden size of $d_{\text{model}} = 2048$, 8 attention heads per layer, and ReLU non-linearities. While GELU activations are common in BERT-style transformers (Devlin et al., 2019), we opt for ReLU due to its more favorable compute-to-throughput ratio (Ming et al., 2022). Each token stream is linearly projected into this shared hidden space, with input dimensionalities being 2048 for both genotypes and phenotypes, and 1024 for fundus image embeddings.

Input sequences are composed of 22 chromosome-level genotype embeddings, a single phenotype token, and two image tokens (left and right eye), with respective sequence lengths of 22, 1, and 2. A learnable positional embedding (Gehring et al., 2017) with values initialized from a truncated normal distribution $\mathcal{N}(0, 0.02^2)$, where values are drawn from a normal distribution centered at zero with a standard deviation of 0.02 and clipped to remain within two standard deviations. This initialization ensures stable early training dynamics while enabling the model to capture relative positional dependencies, particularly important for modalities with inherent structure such as SNP ordering along chromosomes or bilateral symmetry in fundus images. We prepend a learnable [CLS] token and optionally mask selected tokens as part of the training objective. Input corruption is implemented through Gaussian noise (standard deviation $\sigma = 0.08$) and token dropout ($p = 0.4$), reflecting a denoising autoencoder framework and aligning with the strategies proposed in (Vincent et al., 2008). Token corruption is sampled from a categorical distribution over four modes: no corruption (20% probability), additive Gaussian noise (25%), token dropout (25%), and combined noise + dropout (30%). This heterogeneously noised input encourages the model to learn robust representations under varying perturbation levels. In parallel, a token-level masking strategy is applied independently to each modality with a uniform masking probability of 0.5. Masked tokens are replaced with a shared learnable [MASK] vector, as in (He et al., 2022), and the model is trained to reconstruct their clean embeddings through $\mathcal{L}_{\text{mask}}$.

The total training objective is a weighted sum of three terms: masked token reconstruction loss $\mathcal{L}_{\text{mask}}$, denoising loss $\mathcal{L}_{\text{recon}}$, and a cross-modal contrastive loss $\mathcal{L}_{\text{InfoNCE}}$ as detailed in the main text. Each loss term contributes equally to the optimization signal. The contrastive loss is scaled by a factor $\lambda = 0.5$ and uses a softmax temperature $\tau = 0.03$.

Training proceeds for 25 epochs with a batch size of 128 using the quasi-hyperbolic variant of the Adam optimizer (Ma & Yarats, 2018). We use a base learning rate of $5 \cdot 10^{-5}$ and a weight decay of $10^{-4}$. The optimizer's quasi-hyperbolic momentum terms are set to $\nu_1 = 0.7$ and $\nu_2 = 1.0$. A cosine learning rate schedule is used with 10 epochs of linear warm-up starting from 10% of the base rate. A learning rate floor of $10^{-6}$ is imposed to avoid collapse. While we considered dynamic learning rate adjustment via plateau detection, the cosine scheduler offered smoother convergence for our setup. To prevent exploding gradients, we enforce max-norm gradient clipping with a threshold of 10. Mixed precision training was explored but ultimately disabled due to loss gradient instability when scaling with batch size. However, we anticipate that future training runs could benefit from mixed precision.

All experiments were monitored continuously using logging tools. Checkpoints were saved at every epoch, including full model weights, optimizer states, and metrics (refer to Supplementary Fig. 14 for validation curves using different fusion model configurations). Training was conducted on a compute cluster equipped with NVIDIA A100 and A5500 GPUs and 150 GB RAM for data loading and preprocessing. For the full fusion model, a single A100 was used per training run. Validation was performed on a held-out stratified 10% split of the data.

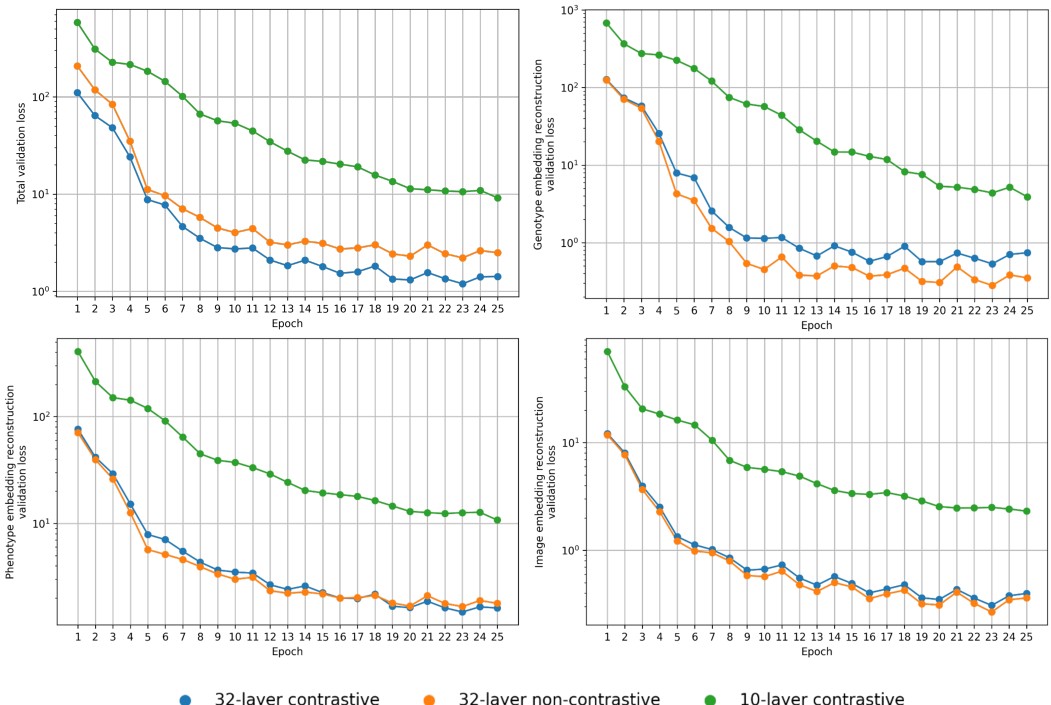

Supplementary Figure 14: **Validation loss curves across model variants.** Total and modality-specific reconstruction validation losses are shown over training epochs for three fusion model configurations: a 32-layer transformer with contrastive loss (blue), a 32-layer transformer without contrastive loss (orange), and a smaller 10-layer contrastive model (green). The top-left panel shows overall validation loss, while the remaining plots show per-modality losses for genotype, phenotype, and image embeddings, respectively. The 32-layer contrastive model consistently achieves the lowest total loss, suggesting better overall integration across modalities.

## G    BENCHMARK SETUP

### G.1    PHENOTYPE MASKING

Supplementary Table 5 shows a taxonomy of target phenotypes including their corresponding related phenotypes that were masked during prediction to prevent label leakage. Each row shows the target phenotype, a brief description, and the set of semantically or clinically related traits that were excluded from the input.

### G.2    MEDGEMMA

MedGemma (Sellergren et al., 2025), is a checkpoint of the multimodal Gemma 3 (Team et al., 2025) foundation model fine-tuned for medical applications. Relative to the base Gemini model, MedGemma benefits from continued pretraining on a broad internal medical corpus, including retinal fundus images, dermatology images across 210 skin conditions, histopathology patch-text pairs, and 2D radiology slices (Team et al., 2025). All following benchmarks are conducted with the largest available 27 billion parameter multimodal MedGemma checkpoint to ensure a fair comparison.

We benchmark MedGemma on phenotype prediction (ROC-AUC) on a representative subset of 1,000 samples from our test set across 9 diseases and systemic conditions. We run inference using the HuggingFace checkpoint (Sellergren et al., 2025) on 2 A100 GPUs (80 GB VRAM). With an average runtime of 64.4 s per sample across nine phenotypes, inference on the full test set of 52,127 participants would require $\approx 8,392$ hours ($\approx 1$ year). Given the strong class imbalance (on average only 739.7 cases per phenotype in the test set, see Supplementary Tables 3 and 4), we instead evaluate on a representative 1,000-participant subset per phenotype, containing all (or up to 500)

Supplementary Table 5: Taxonomy of target phenotypes including their corresponding related phenotypes that were masked during prediction to prevent label leakage.

| Target Phenotype | Description | Masked Phenotypes |
|---|---|---|
| Macular degeneration | A progressive eye disease that causes vision loss in the center of the visual field. | Macular degeneration; Which eye(s) affected by macular degeneration (Right eye); Which eye(s) affected by macular degeneration (Left eye); Eye problems/disorders Macular degeneration; Age macular degeneration diagnosed |
| Diabetic eye disease | Eye damage resulting from diabetes, including retinopathy and macular edema. | Diabetic eye disease; Which eye(s) affected by diabetes-related eye disease (Right eye); Which eye(s) affected by diabetes-related eye disease (Left eye); Eye problems/disorders Diabetic eye disease; Age when diabetes-related eye disease diagnosed |
| Eye problems/disorders Glaucoma | A group of eye conditions that damage the optic nerve, often due to high intraocular pressure. | Glaucoma; Eye problems/disorders Glaucoma; Ever had surgery for glaucoma or high eye pressure; Ever had laser treatment for glaucoma or high eye pressure; Which eye(s) affected by glaucoma (Right eye); Which eye(s) affected by glaucoma (Left eye); TTE glaucoma; Age glaucoma diagnosed; Retinal problem; Retinal detachment; Retinal artery/vein occlusion; TTE chorioretinal inflammation; TTE retinal detachments and breaks; TTE retinal vascular occlusions; TTE other retinal disorders; TTE retinal disorders in diseases classified elsewhere |
| Eye problems/disorders Cataract | Clouding of the lens in the eye leading to a decrease in vision. | Cataract; Eye problems/disorders Cataract; Ever had cataract surgery; Which eye(s) are affected by cataract (Right eye); Which eye(s) are affected by cataract (Left eye); TTE senile cataract; TTE other cataract; Age cataract diagnosed |
| Heart failure | A chronic condition where the heart doesn't pump blood as well as it should. | Heart failure/pulmonary oedema; TTE heart failure |
| AD ischaemic stroke | A type of stroke caused by an obstruction within a blood vessel supplying blood to the brain. | AD ischaemic stroke; Transient ischaemic attack (TIA); TTE transient cerebral ischaemic attacks and related syndromes; TTE cerebral infarction; TTE occlusion and stenosis of precerebral arteries, not resulting in cerebral infarction; TTE stroke, not specified as haemorrhage or infarction; Age stroke diagnosed; TTE other acute ischaemic heart diseases; TTE chronic ischaemic heart disease |
| Heart attack | Also known as myocardial infarction, occurs when blood flow to the heart is blocked. | Heart attack/myocardial infarction; AD myocardial infarction; Vascular/heart problems diagnosed by doctor Heart attack; TTE acute myocardial infarction; TTE subsequent myocardial infarction; Age heart attack diagnosed; TTE other acute ischaemic heart diseases; TTE chronic ischaemic heart disease |
| TTE dementia in Alzheimer's disease | A dementia subtype caused by Alzheimer's disease pathology. | AD Alzheimer's disease; Alzheimer's disease/dementia; TTE dementia in Alzheimer's disease; TTE Alzheimer's disease |
| Alzheimer's disease | A progressive neurodegenerative disorder and the most common cause of dementia. | AD Alzheimer's disease; Alzheimer's disease/dementia; TTE dementia in Alzheimer's disease; TTE Alzheimer's disease; AD all cause parkinsonism; AD Parkinson's disease |

cases and the remainder controls. Since metrics such as ROC-AUC depend on ranking rather than prevalence, this provides a reasonable yet representative simplification that reduces compute time to $\approx 161$ hours.

For phenotype-to-phenotype prediction (P→P), we curate a natural language prompt from the input phenotypes per sample, masking any traits related to the target according to the masking protocol detailed in **Supplementary Section G.1**, and instruct the model to predict the probability of the sample to be be diagnosed with the target phenotype. We subsequently extract the prediction from the answer string and compute an ROC-AUC score. For the multimodal setting (P+I→P), we also included fundus images of both eyes, resized and patched together into a single 896×896 image and normalized to the [–1, 1] range, as required by MedGemma, in the prompt. Again, the model was then prompted to produce a probability reflecting the likelihood that the individual is diagnosed with the target phenotype.

### G.3 CONTIG

We pretrain ContIG following Taleb et al. (2022) using the UK Biobank raw SNP data and retinal fundus images. During pretraining, the model maximizes agreement between image and genetic representations using a contrastive loss. For downstream evaluation, we fine-tune the pretrained encoder on each phenotype prediction task.

