# OpenReview forum: "PM1: A Foundation Model Fusing Genotype, Phenotype, and Image for Precision Medicine"
_ICLR.cc/2026/Conference — ICLR 2026 Conference Withdrawn Submission_

### Official Review · Reviewer_g6Gu · 2025-10-27

**Soundness:** 3
**Presentation:** 4
**Contribution:** 1
**Rating:** 2
**Confidence:** 5

**Summary:**

This paper introduces PM1, a large-scale multimodal foundation model that integrates genotype, phenotype, and retinal image data derived from biobank resources. The model employs a two-stage training pipeline: modality-specific encoders/decoders are trained first, followed by a fusion transformer trained with masked modeling and contrastive losses. The proposed approach achieves strong performance on cross-modal inference, phenotype prediction, and conditional image generation, outperforming several competitive baselines.

**Strengths:**

1. The paper addresses an ambitious and high-impact goal: integrating three complex and heterogeneous data modalities for precision medicine.
2. The token-level masking mechanism for handling missing modalities is a well-designed and practical solution, especially valuable in real-world clinical contexts.
3. The evaluation is exceptionally comprehensive, spanning predictive modeling, cross-modal inference, generative tasks, and biological plausibility assessments (e.g., GWAS and ancestry analyses).

**Weaknesses:**

1. Limited algorithmic novelty: The primary limitation lies in the absence of a fundamentally new algorithmic or theoretical contribution. The proposed architecture combines established components (VAEs, Transformers, InfoNCE) in a competent but largely incremental way. The contribution is primarily one of engineering, scale, and empirical demonstration rather than a conceptual advance.
2. Potential misalignment with ICLR’s scope: The work, while impressive in scope and technical execution, feels more aligned with application-focused venues in biomedical AI than with ICLR’s focus on core machine learning innovation.
3. Nature of the performance gains: The observed performance improvements appear to stem mainly from model scale and comprehensive data integration, rather than from a novel learning component that would generalize to other domains or modalities.

**Questions:**

1. The chromosome-specific VAE compresses each chromosome into a single latent vector. Was any consideration given to more expressive architectures that preserve finer-grained intra-chromosomal representations or functional substructures?
2. Beyond the qualitative t-SNE visualization, could the authors provide a quantitative ablation demonstrating how much the InfoNCE loss contributes to final performance on key prediction tasks?

---

### Official Review · Reviewer_JBxD · 2025-11-01

**Soundness:** 4
**Presentation:** 3
**Contribution:** 4
**Rating:** 8
**Confidence:** 4

**Summary:**

This paper introduces PM1, a foundation model for precision medicine that fuses three complex modalities: genotype (SNPs), phenotype (clinical traits), and image (retinal fundus photos) from hundreds of thousands of patients. Its architecture uses modality-specific encoders (a VAE for genomics, a Masking AE for traits, and RETFound for images) that project data into a shared space. A central fusion transformer is then trained to align these representations using a combined masked-token modeling and InfoNCE contrastive loss. This two-stage approach allows the model to be trained on the full dataset despite massive data missingness (only 6% of participants have all modalities) , and it achieves state-of-the-art performance in downstream tasks like disease prediction and cross-modal synthesis.

**Strengths:**

1. The paper tackles the highly complex challenge of fusing three "difficult" and high-dimensional modalities (genomics, clinical data, and images) at a scale of hundreds of thousands of patients. This is a significant scientific and engineering achievement.
2. The training strategy is a key strength. By using masked token modeling, the model can leverage the entire dataset, not just the 6% of complete samples. The InfoNCE loss (Eq. 2) is perfectly suited to this task, and Figure 2 proves it is essential for aligning the modalities .
3. The model achieves massive performance gains. In Table 1, PM1 outperforms MedGemma by an average 51% relative gain in ROC-AUC . Beating a strong, recent baseline by such a margin is a major result.
4. The paper validates the model beyond simple prediction. The generative tasks (retinal image synthesis, genotype reconstruction) and the group-level probes (GWAS recovery of HERC2, ancestry classification) confirm that the learned latent space is not just predictive but also biologically and semantically meaningful

**Weaknesses:**

1. The generative claim is slightly oversold. The image decoder is not part of the end-to-end model. Image synthesis requires training a separate U-Net/diffusion model that is conditioned on the frozen PM1 embeddings . This is a post-hoc, decoupled process, making PM1 more of a unified representation model than a unified generative model.
2. The two-stage training approach, while computationally necessary, is a limitation (as acknowledged by the authors ). The modality encoders are frozen and cannot be fine-tuned during the fusion stage, meaning their representations cannot adapt to the cross-modal signals being learned by the transformer.
3. The model's foundation is the UK Biobank, which is 89.2% White-British. The authors state this as a limitation and discuss it extensively in the ethics appendix . This is a major, unavoidable weakness that limits the model's generalizability and risks amplifying health disparities.
4. The comparison with MedGemma in Table 1, while showing impressive gains, was run on a 1,000-sample subset because the baseline was too slow for the full test set (est. "≈ 1 year"). This is an asymmetric evaluation. The comparison against ContIG, which was run on the full test set and where PM1 also wins, is a more fair and robust benchmark.

**Questions:**

1. Given that image generation requires training a separate diffusion model conditioned on PM1's embeddings, would it be more accurate to frame PM1 as a unified multimodal representation model rather than a unified generative model?
2. The two-stage training (frozen encoders) is noted as a limitation. Have you explored any parameter-efficient fine-tuning (PEFT) methods, such as LoRA (which you used for the RETFound adaptation ), to allow for jointly fine-tuning the encoders during the fusion stage?
3. Given the severe and acknowledged ancestry bias of the UKB, have you performed a stratified evaluation of the model's phenotype prediction (Table 1) on the non-European cohorts (e.g., African, South-Asian)? Quantifying the performance drop would be a powerful way to underscore this limitation.
4. The SOTA gains over MedGemma are striking. Are you confident the 1,000-sample subset is fully representative and that the performance gap isn't partially an artifact of this small-sample evaluation? (The strong performance against ContIG on the full test set does help alleviate this concern).

---

### Official Review · Reviewer_AjS9 · 2025-11-02

**Soundness:** 2
**Presentation:** 2
**Contribution:** 2
**Rating:** 2
**Confidence:** 4

**Summary:**

PM1 is a tri-modal foundation model (genotype G, phenotype P, fundus image I). It fuses modality-specific encoders (RETFound for images) with a transformer fusion block trained by an InfoNCE-style objective and adds generative decoders for cross-modal reconstruction. The main quantitative result is phenotype prediction (ROC-AUC) where PM1 is compared to MedGemma on a 1,000-sample UKB subset and to ContIG on the full test set; the paper also shows cross-modal genotype inference curves and a GWAS on image-conditioned fusion embeddings recovering HERC2 pigmentation signals.

**Strengths:**

1. Objectives and tasks are described clearly, cross-modal genotype inference across autosomes with multiple input combinations, and downstream phenotype prediction with masking out related traits to reduce leakage.

2. good and clear visualization.

**Weaknesses:**

1. Baselines & fairness are insufficient. The core claims hinge on tri-modal fusion, yet the paper lacks (i) image-only controls built on the same encoder (e.g., RETFound-frozen + linear/MLP per trait), (ii) G+P baselines (e.g., PRS + clinical risk), and (iii) capacity-matched controls (late-fusion MLP vs. the fusion transformer; swap RETFound for a vanilla ViT). The MedGemma comparison covers only P→P and P+I→P, excludes genotype, and is restricted to a 1k subset with prompt-engineering—useful but not a primary yardstick. Action: add RETFound and PRS+clinical baselines, parameter-matched ablations, and broaden MedGemma (smaller checkpoint/batched inference).

2. Quantitative rigor is below ICLR bar outside Table 1. Reports emphasize ROC-AUC without 95% CIs, DeLong tests, or PR-AUC for imbalanced traits; several findings rely on visualizations (t-SNE/PCA, bar plots) rather than statistical comparisons. Action: add CIs and significance tests for all key results; report PR-AUC and calibration (ECE/Brier), especially for the 1k MedGemma subset.

3. Cross-modal genotype inference lacks competitive baselines and diagnostics. Evidence is mainly within-model ablations (P/I/P+I/G+P+I) with per-chromosome plots in the supplement; no comparison to locus-specific predictors, PRS, or simple autoencoder/VQ baselines. Action: add external comparators, per-chromosome ROC/PR with CIs, and calibration analyses to support claims of robust G-inference.

4. GWAS & leakage controls need stronger methodology in the main text. The GWAS on fusion embeddings recovering HERC2 is a good sanity check, but covariates (age/sex/batch/ancestry PCs), multiple-testing procedures, and negative controls are not fully documented. Leakage mitigation via “related-phenotype masking” lives in the supplement and is not stress-tested. Action: move covariates/multiplicity and mask lists into the main paper; add shuffled-embedding/orthogonal-trait controls and report additional reproducible loci beyond HERC2.

5. Generalization and generative utility are under-validated. Clinical prediction is UKB-centric; EyePACS is not used as a labeled external test, and ancestry results appear only as a probing task (risk of population confounding). For image/SNP generation, there’s no clinician blind-grading, lesion-level metrics, OOD tests, or comparisons to training on matched real data or image-only conditioning.

**Questions:**

1. The paper positions PM1 as a unified G–P–I foundation model; however, the strongest quantitative claim (Table 1) is on phenotype prediction, with the MedGemma comparison constrained to a subset and the genotype dimension lacking head-to-head baselines. Tightening the link between tri-modal novelty and tri-modal benchmarks would strengthen the narrative. (Table 1 coverage and comparison framing.

2. Could you report per-chromosome or locus-level accuracy/PR curves and calibration, and compare against standard PRS or locus-specific predictors?

3. What are the full covariate set and multiple-testing procedures used, and include negative controls (e.g., shuffled embeddings)? Also, beyond HERC2, which additional loci reproducibly emerge?

---

### Official Review · Reviewer_G41J · 2025-11-02

**Soundness:** 2
**Presentation:** 3
**Contribution:** 3
**Rating:** 4
**Confidence:** 2

**Summary:**

PM1 proposes a multimodal foundation model for genotype (G), phenotype/tabular (P, 3,421 features), and retinal images (I). It uses modality-specific encoders (genotype CNN-VAE, phenotype DAE, image MAE) and a fusion Transformer trained with masked reconstruction (imputation-style) + InfoNCE. Tasks include phenotype prediction (mask→reconstruct), cross-modal genotype inference (e.g., P/I→G), and conditional retinal image generation via a separate diffusion model. Results show strong gains over two multimodal baselines (MedGemma, ContIG) and internal ablations.

**Strengths:**

1. **Practical multimodal integration across highly heterogeneous data (G/P/I)**
The model integrates genotype, phenotype, and image data using a reconstruction-based interface that naturally accommodates missing modalities.

2. **Clear, scalable engineering**
The architecture cleanly separates modality-specific encoders from a shared fusion Transformer, allowing for easy scaling and future extension.

3. **Strong empirical performance on several clinically relevant endpoints**
The method shows consistent improvements in disease classification tasks across diverse modality combinations, validating its utility.

4. **Dataset scale and realism**
Training on large biobank-scale data with real-world missingness makes the model robust and applicable to practical clinical scenarios.

5. **Reproducibility**
The paper provides sufficient architectural and training details to support reproducibility and downstream adaptation by others.

**Weaknesses:**

1. **Unfair and Non-Equivalent Baselines**
The external baselines (MedGemma and ContIG) are not methodologically comparable to PM1.

- MedGemma is a vision–language model evaluated in zero-shot prompt mode without fine-tuning, using textified tabular inputs (P) and images (I).

- ContIG is a genotype–image contrastive learner with a supervised classification head but no phenotype (P) modality.

Thus, the baselines differ from PM1 in available modalities and training objectives, making the numerical comparison not directly fair.

2. **Missing Traditional Tabular Baselines**
The paper does not compare PM1 against strong tabular baselines such as LightGBM, CatBoost, or FT-Transformer for phenotype-only (P→P) prediction. Without these comparisons, it is unclear whether the reported gains genuinely come from multimodal fusion or simply from the absence of optimized tabular models on the 3,421 phenotype features.

3. **Large and Unexplained Performance Gap**
PM1 outperforms both MedGemma and ContIG by a very large margin (up to +50 AUC points in some tasks), yet the paper provides no systematic analysis explaining why.

4. **ContIG’s Missing Modality Makes the Comparison Unbalanced**
ContIG models only images and genotypes (I↔G), while PM1 additionally uses phenotype tables (P).
Comparing PM1 (which accesses richer information) against ContIG might conflate architectural improvements with data advantage.
A fair comparison would either restrict PM1 to the same (I, G) inputs or extend ContIG with a lightweight tabular adapter to include P.

**Questions:**

1. **Baseline selection & comparability**
Could the authors clarify how the multimodal baselines were selected and whether all methods were trained/evaluated under comparable settings—in particular, with the same amount of information (matched input modalities)?

2. **MedGemma dataset**
Is MedGemma trained or fine-tuned on the same exact dataset/splits as PM1, or is it evaluated purely in zero-shot prompt mode?

3. **Scope of ContIG fine-tuning**
The supplement said ContIG is fine-tuned “on phenotype prediction tasks.” Does this mean predicting all 3,421 phenotype features, or only the nine disease endpoints reported in Table 1 (e.g., macular degeneration, glaucoma, etc.)?


4. **Use of P (3,421 features) in baselines**
Table 1 suggests P is highly informative. Is the same P (3,421 tabular features) used—in its native tabular form—to train any comparison method such as MedGemma?


5. **Explaining the large performance gap**
The reported AUC gains over MedGemma and ContIG are very large (up to +50 points). Could the authors provide an explanation to confirm the gap is not due to scale or data advantage?

6. **The t-SNE Figure (Fig. 2)**
What is the criterion for “good” in Figure 2? Is it that the left and right eyes are well mixed in the latent space? Should we expect the phenotype features (yellow) to be mixed with genetic and image features?

7. **Image Generation**
How was the image generation performance evaluated? Are there any quantitative comparisons with other methods? The generated images in Figure 3 seem blurry — is this expected?

8. **About Genotype Generation**
What is the purpose of generating proxy gene embeddings? How is this useful in practice or clinical application? Are the generated embeddings ever validated or used for downstream predictions?

---

### Note · Authors · 2025-12-02

I have read and agree with the venue's withdrawal policy on behalf of myself and my co-authors.